# Exploring hypoxia-related genes as prognostic indicators in lung adenocarcinoma

**Bandar Alghamdi[1], Sonia Rocha[2]\***

**1** King Fahad Army Forces Hospital, Cardiology and Cardiac Surgery Center, Jeddah, Saudi Arabia,
**2** Department of Biochemistry, Cell and Systems Biology, Institute of Systems Molecular and Integrative Biology, University of Liverpool, Liverpool, United Kingdom

\* srocha@liverpool.ac.uk

## Abstract

### Background

Lung adenocarcinoma (LUAD) is a leading cause of cancer-related mortality, with hypoxia contributing to tumor progression and treatment resistance. Identifying hypoxia-related biomarkers could enhance prognosis and therapeutic strategies for LUAD.

### Methods

This study aimed to identify hypoxia-related differentially expressed genes (HRDEGs) in lung adenocarcinoma (LUAD) through differential expression analysis. Functional analysis and protein-protein interaction (PPI) network construction were performed to explore the biological roles and interactions of these genes. Kaplan-Meier survival analysis, univariate Cox regression, and Lasso regression were used to identify key genes associated with survival. Multivariate Cox regression was then conducted to assess independent prognostic factors.

### Results

This analysis revealed 283 upregulated HRDEGs and 322 downregulated HRDEGs in LUAD. Functional enrichment analysis indicated that the upregulated genes were primarily involved in cancer-related and cellular signaling pathways, while downregulated genes were associated with immunity-related pathways. We further identified 201 common upregulated hub genes (including *MMP9, CDH1, HSP90AB1, SOX2, CDKN2A, SPP1, EZH2*) and 224 common downregulated hub genes (such as *IL6, TNF, IL1B, JUN, CCL2, TLR4, FOS, PTGS2*). Kaplan-Meier survival analysis, univariate Cox regression, and Lasso regression led to the identification of 17 key genes (*ADRB2, ALDH2, CAT, CCNE1, MAP3K8, DSG2, EIF6, ABCB1, PIK3R1, RAD51, SFTPD, SOD3, CLEC3B, ADAM12, EXO1, FBLN5,* and *IGF2BP3*) associated with patient survival. Finally, multivariate Cox regression analysis identified *DSG2, EIF6,*

**Data availability statement:** All relevant data are publicly available. The transcriptomic and genomic datasets analyzed in this study were retrieved from publicly accessible repositories, including GEO (Gene Expression Omnibus) and TCGA (The Cancer Genome Atlas). Specific accession numbers and dataset identifiers are provided within the manuscript. Accession numbers and links are: GSE18842; GSE13213; https://www.cbioportal.org/study/summary?id=luad_tcga. No new data were generated.

**Funding:** This study was supported by the Wellcome Trust to SR (206293/Z/17/Z). The funders had no role in study design, data collection and analysis, decision to publish, or preparation of the manuscript.

**Competing interests:** The authors have declared that no competing interests exist.

and *EXO1* as independent prognostic factors for LUAD, highlighting their potential as biomarkers for prognosis and therapeutic targets in lung cancer.

## Conclusion

In conclusion, *DSG2, EIF6,* and *EXO1* were identified as key hypoxia-related genes in lung adenocarcinoma. These genes were found to be independent prognostic factors, highlighting their potential as biomarkers for predicting patient survival and guiding future therapeutic approaches.

## Introduction

Lung Adenocarcinoma and Hypoxia Lung cancer remains one of the most common and deadly cancers worldwide, with non-small cell lung cancer (NSCLC) being the dominant histological subtype [1]. Lung adenocarcinoma (LUAD) is the most prevalent histological subtype of NSCLC and represents a major clinical and biological focus within this disease category due to its distinct molecular and genetic characteristics. Although advances in molecular profiling and targeted therapies have improved the management of NSCLC, the prognosis for patients with LUAD remains poor, particularly in advanced stages [2]. The overall five-year survival rate for LUAD patients is low, making it essential to identify novel biomarkers for early diagnosis, prognosis, and therapeutic targeting [3]. One of the critical microenvironments influencing cancer progression, metastasis, and resistance to therapy is tumor hypoxia. Hypoxia refers to the state of low oxygen availability within the tumor, often resulting from an imbalance between the rapid growth of tumor cells and the insufficient development of blood vessels to supply oxygen [4]. Hypoxia is a hallmark feature of many solid tumors, including lung adenocarcinoma, and it is associated with aggressive tumor behavior, poor prognosis, and resistance to conventional therapies [5]. The development of therapeutic strategies targeting hypoxic tumor cells remains an area of active research.

Hypoxia plays a crucial role in driving various aspects of tumor biology, such as tumor angiogenesis, metabolic reprogramming, and epithelial-to-mesenchymal transition (EMT) [6]. It activates a range of adaptive cellular responses through hypoxia-inducible factors (HIFs), primarily HIF-1α and HIF-2α [7]. These transcription factors regulate the expression of genes that mediate critical processes such as angiogenesis (formation of new blood vessels), glycolysis, metastasis, and drug resistance [8,9]. Tumor cells exposed to hypoxic conditions undergo metabolic shifts that favor anaerobic glycolysis, which allows them to survive and proliferate despite low oxygen availability. The HIF signaling pathway is essential for the regulation of genes involved in cellular survival, metabolism, and tumor progression [8,10]. Among the most well-known hypoxia-regulated genes are vascular endothelial growth factor (VEGF), glucose transporter proteins (e.g., GLUT1), and matrix metalloproteinases (e.g., MMPs) [11]. These genes promote tumor growth and metastasis by facilitating the adaptation of tumor cells to low-oxygen conditions [12–14]. Additionally, hypoxia

is implicated in the regulation of immune evasion and resistance to chemotherapies and radiation therapy, further complicating treatment strategies [15]. Therefore, the identification of key hypoxia-related genes and their role in LUAD progression and prognosis is crucial for improving the clinical management of lung adenocarcinoma.

The link between hypoxia and poor prognosis in LUAD has been well established through various studies. Tumors with hypoxic regions have been shown to have a higher potential for metastasis, increased tumor aggressiveness, and reduced sensitivity to treatment [16]. Patients with high expression levels of hypoxia-related genes often exhibit shorter overall survival and progression-free survival compared to those with low expression levels of these genes in many cancers [17]. Several studies have sought to identify specific hypoxia-related biomarkers that could serve as predictors of patient outcomes in LUAD [18–20]. Li et al. analyzed immune–hypoxia-related immune genes and developed a 7-gene prognostic signature; while informative, their study focused narrowly on genes intersecting immune and hypoxia pathways, without comprehensive assessment of hub genes or functional interactions [18]. Xiong et al. identified 15 hypoxia-related genes to classify hypoxia clusters and constructed a ceRNA-based prognostic network; however, their analysis prioritized clustering and non-coding RNA networks, with limited evaluation of independent prognostic value of core genes [19]. The Malawi Medical Journal study established a 9-lncRNA hypoxia signature to predict survival, but focused exclusively on non-coding RNAs and lacked broader gene-level validation across multiple cohorts [20]. Collectively, these studies provided valuable insights into hypoxia-driven mechanisms, but none performed a genome-wide identification of hypoxia-related genes, systematically integrated PPI networks, and validated independent prognostic genes across multiple datasets. In contrast, the present study performs a comprehensive, multilayered analysis of hypoxia-related differentially expressed genes, integrating functional enrichment, protein–protein interaction networks, and survival modeling to identify independent prognostic hub genes. This approach not only captures broader hypoxia-induced changes in LUAD but also provides mechanistic insights into key pathways and biological processes, thereby extending the current understanding of hypoxia-driven tumor progression and prognosis.

However, the heterogeneity of LUAD, coupled with the complexity of hypoxia-induced changes in gene expression, presents challenges in pinpointing specific prognostic genes. Additionally, the tumor microenvironment in LUAD is influenced by various factors, such as genetic mutations, immune responses, and stromal interactions, which may confound the impact of hypoxia on survival [21,22]. While some studies focused on a small number of hypoxia-induced genes [19], other studies were interested in immune phenotypes [18,19] and non-coding RNAs [19,20] as potential biomarkers of the disease. These studies, using robust statistics and analysis pipelines produced valid and potential biomarkers in these subareas of hypoxia induced changes, with varying prognostic value. However, they all lacked validation in independent cohort of data. Furthermore, by using prior selection of given criteria such small number of hypoxia-induced genes, focusing mostly on non-coding RNAs, these studies increased biases and also might have lost critical information in other areas of the data available.

In this context, a comprehensive analysis of hypoxia-related genes and their potential prognostic value in LUAD is thus still missing. Identifying genes that are consistently associated with poor prognosis, regardless of other clinical variables, will allow for the development of better prognostic models. These models could guide personalized treatment strategies, such as targeting hypoxic regions of the tumor with hypoxia-activated pro-drugs or improving the efficacy of therapies through the modulation of the HIF pathway.

This study aims to systematically investigate hypoxia-related genes in LUAD and evaluate their potential prognostic significance. We used a combination of computational and statistical approaches to analyze large-scale genomic datasets, while also validating the prognostic value of key findings in an independent cohort of patients. By identifying key hypoxia-associated genes, this work seeks to find independent prognostic biomarkers and provide insights into hypoxia-driven mechanisms underlying disease progression in LUAD Our iterative and multilayered analysis lead to the identification of three genes that held across all datasets analyzed and had predictive prognostic value in LUAD.

## Methods

### Data downloading and processing

The analysis utilized two distinct datasets: the TCGA LUAD cohort and GSE18842. We downloaded the TCGA LUAD cohort form the cBioPortal [23]. The TCGA LUAD cohort comprised 517 tumor samples and 59 control samples, representing lung adenocarcinoma (LUAD) patients and normal controls, respectively (https://www.cbioportal.org/study/summary?id=luad_tcga). Clinical data (https://www.cbioportal.org/study/summary?id=luad_tcga) from this cohort were incorporated for Kaplan-Meier survival analysis, univariate regression analysis, and multivariate regression analysis to investigate the relationship between gene expression and survival outcomes. For the GSE18842 dataset [24], which contains 46 tumor samples and 45 control samples. data normalization was performed using a log2 transformation. To account for multiple probes corresponding to individual genes, the data was averaged, resulting in a single expression value for each gene. The GSE18842 dataset is publicly available and can be accessed at https://www.ncbi.nlm.nih.gov/geo/query/acc.cgi?acc=GSE18842.

### Identification of differentially expressed gene (DEGs) and hypoxia related differentially expressed gene (HRDEFs)

In this study, we utilized multiple datasets to identify differentially expressed genes (DEGs) associated with tumorigenesis. To identify DEGs between tumor and normal samples, we employed the LIMMA package [25], applying a stringent cutoff of $|log2FC| > 0.585$ and an adjusted p-value $< 0.05$. Following DEG identification, we specifically targeted hypoxia-related differentially expressed genes (HRDEGs), which were selected through a Venn diagram approach. This method allowed us to cross-reference the LUAD DEGs and further refine the set of genes related to hypoxia, thus providing a focused analysis of the genes potentially involved in the tumor microenvironment's hypoxic conditions.

### Collection of hypoxia related genes and identification of hypoxia related differentially expressed genes (HRDEGs)

To curate a comprehensive list of hypoxia-related genes, we integrated data from several established resources. First, we retrieved hypoxia-related genes from Gene Set Enrichment Analysis (GSEA) (https://www.gsea-msigdb.org/gsea/index.jsp) [26], focusing on the following gene sets: GO BP: hypoxia inducible factor 1 alpha signaling pathway, GO BP: Intrinsic apoptotic signaling pathway in response to hypoxia, GO BP: Negative regulation of cellular response to hypoxia, GO BP: Negative regulation of hypoxia induced intrinsic apoptotic signaling pathway, GO BP: Regulation of cellular response to hypoxia, hallmark hypoxia, and Reactome: Regulation of gene expression by hypoxia inducible factor. We then performed a search on OMIM (https://www.ncbi.nlm.nih.gov/omim/?term=HYPOXIA) using the term "HYPOXIA" to identify genes associated with hypoxia-related genetic disorders. Additionally, we retrieved genes from the hsa04066:HIF-1 signaling pathway in the KEGG database (https://www.genome.jp/kegg/). Finally, we obtained genes from GeneCards (https://www.genecards.org/), selecting those with a score greater than or equal to 1. After compiling the data from these sources, we performed deduplication to ensure that only unique genes were included. This resulted in a final list of 3292 unique genes, which are tabulated in Supplementary Table S1 in S1 File.

### Functional enrichment analysis of DEGs

We used DAVID v6.8 (https://david.ncifcrf.gov/) [27] to perform functional annotation of the identified genes. This platform allowed us to categorize the genes into various biological contexts, such as significantly enriched KEGG pathways, biological processes, cellular components, and molecular functions. These functional analyses provided valuable insights into the roles and interactions of the genes in key cellular mechanisms. To assess the statistical significance of the enrichment, we applied a false discovery rate (FDR) threshold of < 0.05, calculated using the Benjamini–Hochberg method [28], ensuring that only robust and biologically relevant findings were included.

### Exploration of hub HRDEGs

We constructed Protein-Protein Interaction (PPI) networks for the differentially expressed genes (DEGs) using STRING v11.0 (https://string-db.org/) [29]. Both upregulated and downregulated genes were entered into the STRING search module to generate the interaction networks. To identify key hub genes within these networks, we utilized the cytoHubba plugin in Cytoscape (https://cytoscape.org/) [30], which ranks genes based on their interaction degree. A medium interaction score threshold of ≥0.40 was applied to build the PPI network, and genes with an interaction degree of ≥10 were selected for further analysis. This approach enabled us to focus on the most significant hub genes, highlighting those with strong interactions relevant to lung cancer.

### Exploration of survival associated hub HRDEGs

We retrieved clinical data from TCGA LUAD to perform an analysis of overall survival (OS) in relation to gene expression. Patients were categorized into two groups based on their gene expression levels, either above or below the median, and survival rates between these groups were compared. To further explore survival differences, patients were also stratified into risk groups based on their gene expression profiles. The log-rank test and Kaplan-Meier survival curves were utilized to evaluate and visualize differences in survival outcomes. For a more detailed survival analysis, we used the R package "survival" [31], performing both univariate and multivariate Cox regression analyses to assess the independent effects of genes on survival. Additionally, we applied LASSO-Cox regression via the "glmnet" package [32], incorporating 10-fold cross-validation to optimize the selection of penalty parameters and minimize the risk of overfitting. This comprehensive approach allowed us to identify significant survival-associated genes while controlling for potential biases and overfitting, ensuring robust and reliable findings.

### Validation of prognostic signature in an independent cohort

To validate the prognostic relevance of the identified three-gene signature, we utilized the independent LUAD dataset GSE13213 via the SurvExpress platform [33]. SurvExpress is an online tool that allows assessment of gene expression-based prognostic models across independent cohorts. The multivariate Cox regression model included only the expression levels of these three genes. The expression levels of gene signature were used to calculate a risk score for each patient, and patients were stratified into high-risk and low-risk groups based on the median risk score. Survival differences between the groups were evaluated using Kaplan-Meier survival analysis and the Log-Rank test, and hazard ratios (HRs) with 95% confidence intervals were computed to assess prognostic significance. Additionally, expression differences of gene signature between the risk groups were compared to confirm consistency with the primary cohort findings.

### Statistical analysis and overall methodology

All statistical analyses were performed using R software (version 4.0.1). Differentially expressed genes (DEGs) were identified using a cutoff threshold of $|\log_2$ fold change $(\log_2 FC)| > 0.585$ and an adjusted p-value $< 0.05$. Multiple testing correction was conducted using the Benjamini–Hochberg false discovery rate (FDR) method, with an FDR threshold of $< 0.05$ applied to control for false positives [28]. A p-value threshold of 0.05 was used for the survival analysis of HRDEGs, applied to Kaplan–Meier method, log-rank test, and Cox regression analysis. The overall methodology provided in the Fig 1.

## Results

### Identification of hypoxia related differentially expressed genes (HRDEGs) in Lung adenocarcinoma

We explored differentially expressed genes (DEGs) in the TCGA LUAD cohort (Supplementary Table S2 in S1 File) and identified additional DEGs from the GSE18842 dataset (Supplementary Table S3 in S1 File). Volcano plots depicting the

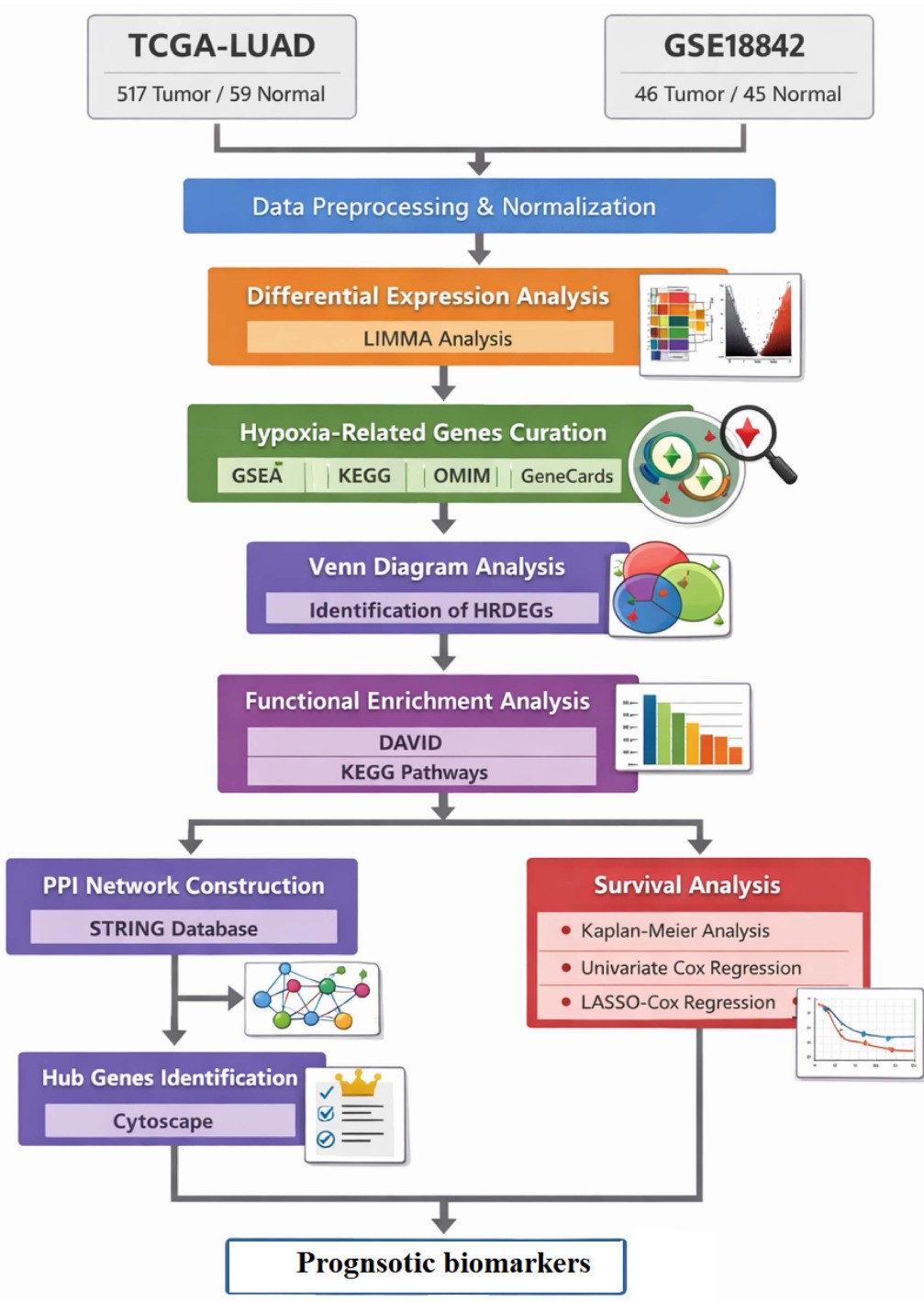

**Fig 1. The overall methodology of the study.**

results of both analyses are shown in Fig 2A and Fig 2B, respectively. These plots provide a visual representation of the relationship between fold changes (logFC) and adjusted p-values, helping to identify genes with significant differential

**Fig 2. Volcano plots and Venn diagrams depicting DEGs and HRDEGs in LUAD. A.** Volcano plot of DEGs in the TCGA LUAD cohort. **B.** Volcano plot of DEGs in the GSE18842 dataset, illustrating differential gene expression in LUAD. **C.** Venn diagram showing the overlap of the upregulated hypoxia related differentially expressed genes (HRDEGs) in LUAD. **D**. Venn diagram illustrating the overlap of downregulated HRDEGs in LUAD.TCGA UP: TCGA-LUAD upregulated; TCGA DOWN: TCGA-LUAD downregulated; HRG: hypoxia-related genes.

expression in lung adenocarcinoma (LUAD). Next, we compared the identified DEGs with hypoxia-related genes to determine the hypoxia-related differentially expressed genes (HRDEGs). This comparison revealed a total of 283 upregulated HRDEGs and 322 downregulated HRDEGs in LUAD, which are displayed in Fig 2C-2D and detailed in Supplementary Table S4 in S1 File. These findings indicate that specific hypoxia-related genes are significantly altered in LUAD, providing potential insights into the molecular mechanisms underlying the disease. The top ten upregulated and downregulated HRDEGs are listed in Table 1. Among the top upregulated HRDEGs are *CYP24A1, CA9, MMP13, SPP1, COL10A1, MMP1, EEF1A2, GREM1, PITX1,* and *MMP12.* These genes are associated with various biological processes such as cell metabolism, matrix remodeling, and tumor progression. On the other hand, the top downregulated HRDEGs include *SFTPC, SLC6A4, AGER, FABP4, SCGB1A1, VEGFD, HBB, SFTPA2, AQP4,* and *CLEC3B*, which are involved in processes like lung surfactant production, cell signaling, and vasculature maintenance.

## HRDEGs are associated with cellular processes such as cell cycle and metabolism

Fig 3 illustrates the pathways significantly enriched by the upregulated genes, with a false discovery rate (FDR) < 0.05. These pathways highlight key biological processes and molecular mechanisms potentially associated with tumorigenesis (Supplementary Table S5 in S1 File). Notably, the cell cycle (hsa04110), biosynthesis of amino acids (hsa01230), and carbon metabolism (hsa01200) are central to cellular growth and energy production. The p53 signaling pathway (hsa04115) and cellular senescence (hsa04218) are also implicated, reflecting the regulation of cell death and aging processes. Other enriched pathways, such as glycolysis/gluconeogenesis (hsa00010) and HIF-1 signaling (hsa04066), suggest alterations

**Table 1. The top ten upregulated and downregulated HRDEGs in TCGA and GSE18842.**

| Regulatory status | Gene ID | Gene Symbol | TCGA | | GSE18842 | | Name of the genes |
|---|---|---|---|---|---|---|---|
| | | | logFC | Adjusted P Value | logFC | Adjusted P Value | |
| Upregulated | 1591 | *CYP24A1* | 5.211 | $8.81 \times 10^{-33}$ | 1.293 | $9.12 \times 10^{-06}$ | cytochrome P450 family 24 subfamily A member 1 |
| | 768 | *CA9* | 4.929 | $2.12 \times 10^{-27}$ | 1.749 | $3.30 \times 10^{-13}$ | carbonic anhydrase 9 |
| | 4322 | *MMP13* | 4.885 | $2.07 \times 10^{-29}$ | 1.825 | $1.41 \times 10^{-07}$ | matrix metallopeptidase 13 |
| | 6696 | *SPP1* | 4.739 | $1.41 \times 10^{-49}$ | 2.540 | $2.92 \times 10^{-20}$ | secreted phosphoprotein 1 |
| | 1300 | *COL10A1* | 4.653 | $6.9 \times 10^{-60}$ | 3.547 | $8.02 \times 10^{-21}$ | collagen type X alpha 1 chain |
| | 4312 | *MMP1* | 4.651 | $1.47 \times 10^{-33}$ | 5.110 | $5.85 \times 10^{-21}$ | matrix metallopeptidase 1 |
| | 1917 | *EEF1A2* | 4.297 | $1.69 \times 10^{-18}$ | 0.989 | $2.14 \times 10^{-05}$ | eukaryotic translation elongation factor 1 alpha 2 |
| | 26585 | *GREM1* | 4.257 | $1.7 \times 10^{-37}$ | 4.666 | $8.94 \times 10^{-24}$ | "gremlin 1, DAN family BMP antagonist" |
| | 5307 | *PITX1* | 4.233 | $9.46 \times 10^{-32}$ | 1.629 | $6.43 \times 10^{-17}$ | paired like homeodomain 1 |
| | 4321 | *MMP12* | 4.085 | $1.72 \times 10^{-29}$ | 5.626 | $7.05 \times 10^{-29}$ | matrix metallopeptidase 12 |
| Downregulated | 6440 | *SFTPC* | −8.560 | $3.4 \times 10^{-38}$ | −6.0378 | $1.01 \times 10^{-25}$ | surfactant protein C |
| | 6532 | *SLC6A4* | −7.971 | $7.8 \times 10^{-90}$ | −1.9304 | $1.49 \times 10^{-15}$ | solute carrier family 6 member 4 |
| | 177 | *AGER* | −6.765 | $2.62 \times 10^{-72}$ | −4.9568 | $1.01 \times 10^{-44}$ | advanced glycosylation end-product specific receptor |
| | 2167 | *FABP4* | −6.160 | $9.71 \times 10^{-77}$ | −3.4366 | $5.77 \times 10^{-26}$ | fatty acid binding protein 4 |
| | 7356 | *SCGB1A1* | −5.786 | $2.46 \times 10^{-22}$ | −3.9076 | $3.06 \times 10^{-08}$ | secretoglobin family 1A member 1 |
| | 2277 | *VEGFD* | −5.008 | $2.11 \times 10^{-46}$ | −4.4324 | $4.79 \times 10^{-40}$ | vascular endothelial growth factor D |
| | 3043 | *HBB* | −4.916 | $1.54 \times 10^{-51}$ | −2.5255 | $1.07 \times 10^{-25}$ | hemoglobin subunit beta |
| | 729238 | *SFTPA2* | −4.863 | $6.09 \times 10^{-23}$ | −3.7907 | $6.99 \times 10^{-10}$ | surfactant protein A2 |
| | 361 | *AQP4* | −4.688 | $1.29 \times 10^{-27}$ | −3.9426 | $9.98 \times 10^{-26}$ | aquaporin 4 |
| | 7123 | *CLEC3B* | −4.241 | $2.27 \times 10^{-72}$ | −4.4159 | $8.41 \times 10^{-38}$ | C-type lectin domain family 3 member B |

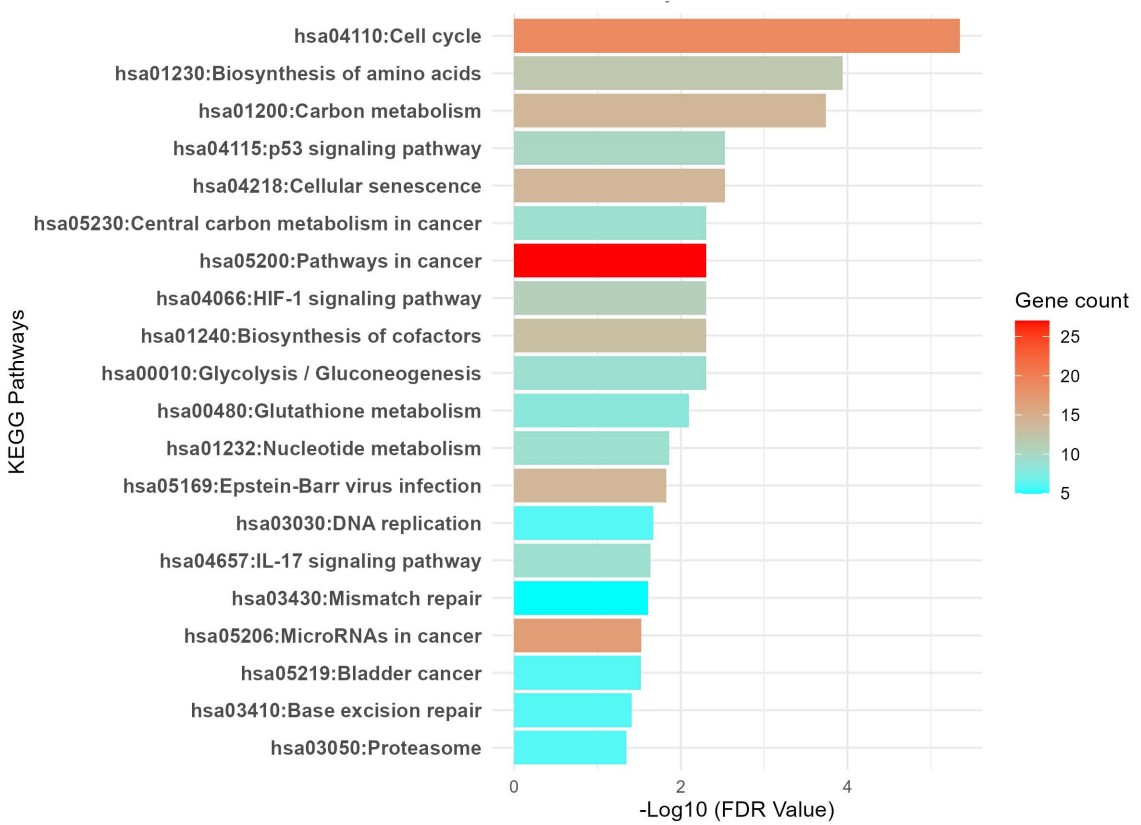

**Fig 3. Significantly enriched pathways associated with the upregulated genes (FDR<0.05).** DEG upregulated were analyzed for functional pathways using DAVID.

in metabolic pathways and hypoxic responses often observed in cancer. Additionally, pathways related to cancer progression, such as central carbon metabolism in cancer (hsa05230), pathways in cancer (hsa05200), and microRNAs in cancer (hsa05206), are also prominently featured, highlighting the complex molecular interactions in cancer biology. Other enriched pathways include DNA replication (hsa03030), mismatch repair (hsa03430), and glutathione metabolism (hsa00480), which are critical for genome stability and oxidative stress responses. The statistical analysis of upregulated pathways is tabulated in Supplementary Table S5 in S1 File. These results emphasize the multifaceted nature of the upregulated genes and their involvement in various cellular functions relevant to cancer development.

The top enriched pathway, the Cell Cycle (Fig 4), with a highly significant p-value of 4.51E-06. This pathway is critical for regulating cell division and proliferation. The genes enriched within this pathway, including *PCNA, MCM7, CDKN2A, PRKDC, HDAC1, CDC6,* and others such as *CDK4, CHEK1,* and *E2F3*, play pivotal roles in cell cycle aberration. These genes are integral to maintaining proper cell cycle regulation, and their upregulation suggests a potential disruption in normal cell division processes, which is commonly associated with tumorigenesis. The enrichment of these genes further underscores the importance of the cell cycle in hypoxic lung cancer biology.

Similar to the Cell Cycle pathway, another significantly enriched pathway is the HIF-1 signaling pathway (hsa04066), with genes such as *LDHA, EGLN3, SLC2A1, EIF4EBP1, PGK1, TIMP1, ENO1, ALDOA, GAPDH, PFKP,* and *PDK1* being upregulated (Fig 5). The upregulation of these genes suggests a critical role of hypoxia in lung cancer, as these genes are involved in metabolic adaptation to low oxygen conditions [34], many of which are known HIF-dependent targets [12].

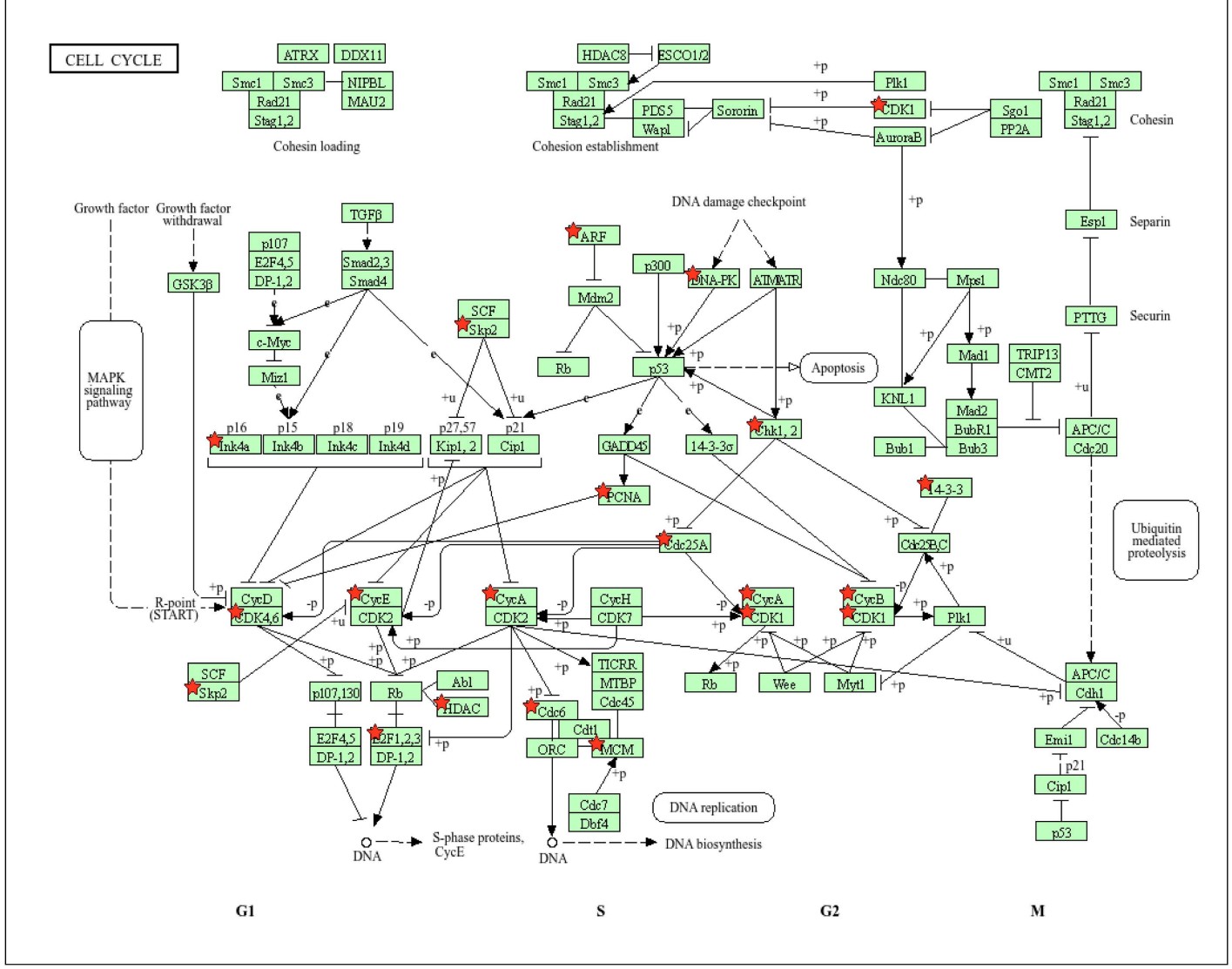

**Fig 4. KEGG pathway illustrating the cell cycle pathway from DAVID, with upregulated genes marked by red stars.**

This highlights the importance of hypoxia-induced signaling in promoting tumor survival and progression through altered cellular metabolism.

In addition to the upregulated genes, we observed that the downregulated genes are predominantly associated with immune-related pathways and signaling processes. As illustrated in Fig 6, several immune-related pathways are significantly affected, indicating a potential suppression of immune responses in lung cancer. Key pathways include the TNF signaling pathway (hsa04668), Cytokine-cytokine receptor interaction (hsa04060), and IL-17 signaling pathway (hsa04657), all of which play essential roles in immune activation and inflammation. Other important pathways such as Toll-like receptor signaling (hsa04620), JAK-STAT signaling (hsa04630), and Natural killer cell-mediated cytotoxicity (hsa04650) are also downregulated, suggesting that the immune system's ability to recognize and eliminate cancer cells

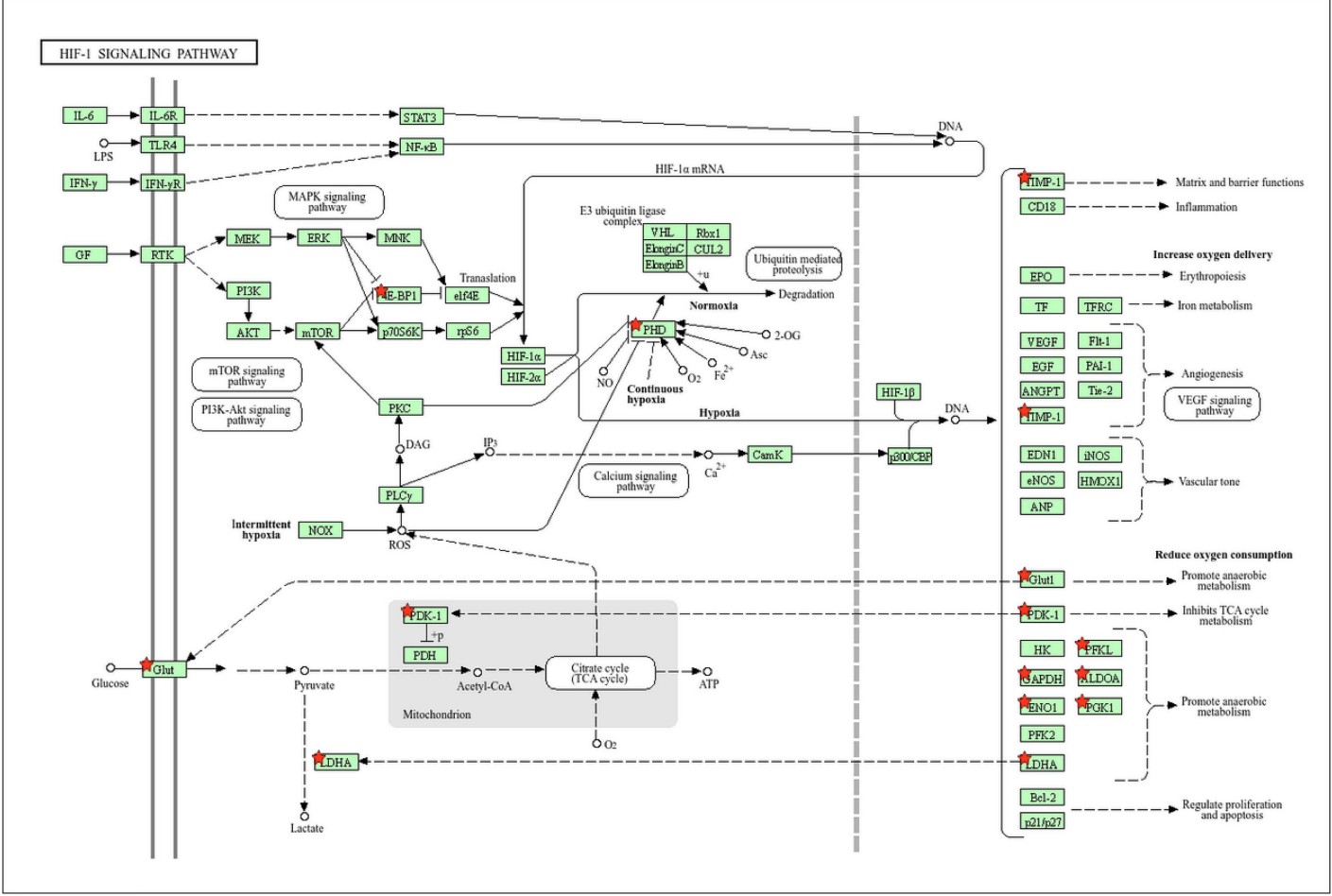

**Fig 5. KEGG pathway illustrating the HIF-1 signaling pathway from DAVID, with upregulated genes marked by red stars.** Upregulated genes, such as *LDHA, EGLN3, SLC2A1, EIF4EBP1, PGK1, TIMP1, ENO1, ALDOA, GAPDH, PFKP, and PDK1*, are shown with red stars, indicating their involvement in metabolic reprogramming and cellular adaptation to low oxygen conditions.

may be impaired. The statistical analysis of upregulated pathways is tabulated in Supplementary Table S6 in S1 File. These findings suggest a potential immune evasion mechanism in lung cancer, where the downregulation of key immune pathways may allow tumors to escape immune surveillance and progress unchecked.

## Explorations of hub HRDEGs

Next, we conducted a Protein-Protein Interaction (PPI) analysis by inputting all the hypoxia-related differentially expressed genes (DEGs) into the STRING database. The primary PPI network consisted of 600 nodes and 9,281 edges, with an average node degree of 30.9 (Fig 7). The PPI enrichment p-value was found to be < 1.0e-16, indicating a highly significant interaction among the hypoxia-related genes. Based on a degree of interaction greater than 10, we identified the hub genes within the network. After identifying the hub genes based on the degree of interaction, we found that the top hub genes include *GAPDH, IL6, TNF, IL1B, MMP9, JUN, CCL2, TLR4, FOS, PTGS2, PECAM1, CDH1, HSP90AB1, PTPRC, ICAM1, FGF2, PPARG, ITGAM, CXCL12, and KDR.* The list of all interacted genes and their degree of interaction provided in the Supplementary Table S7 in S1 File. These genes play pivotal roles in key biological processes such

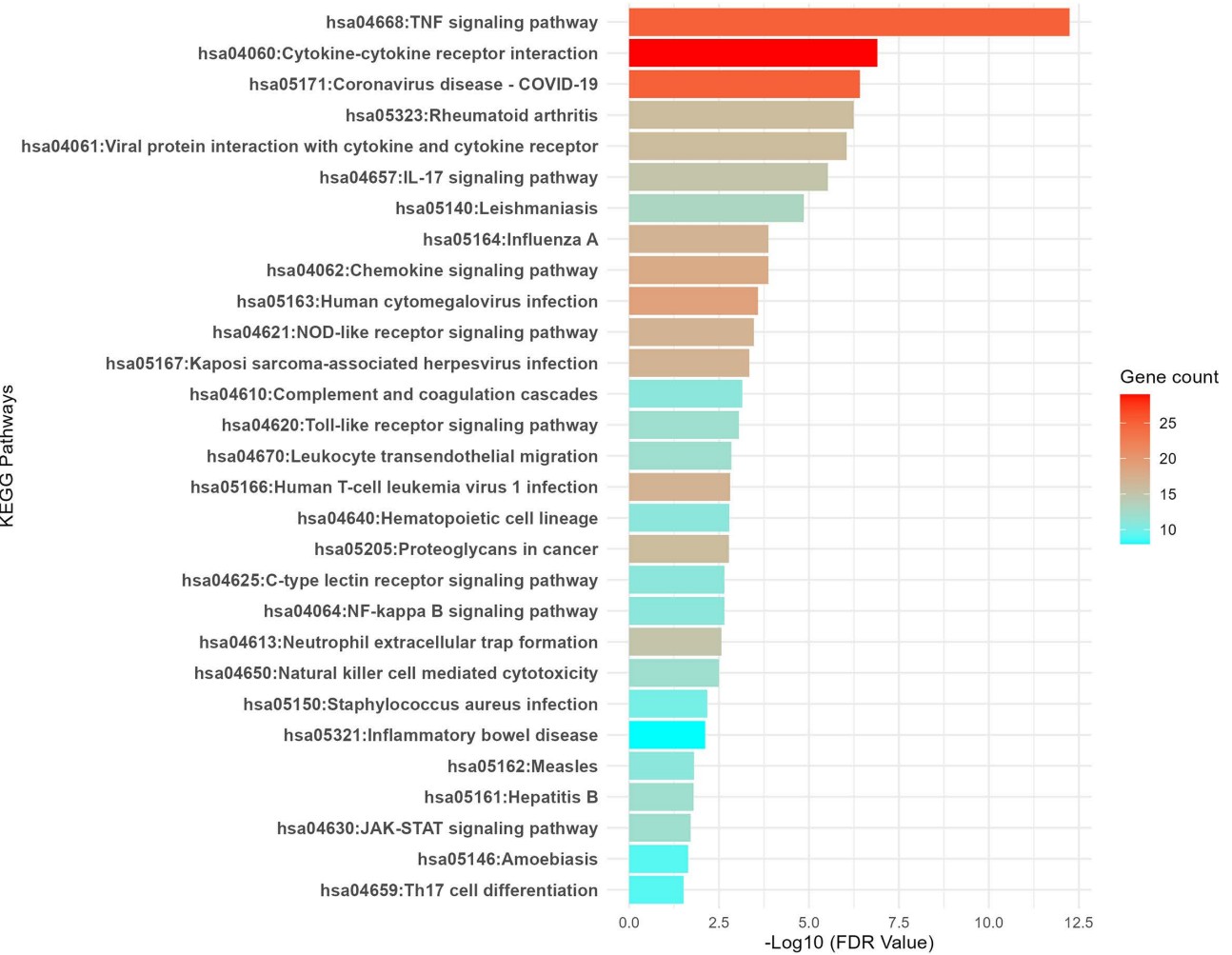

**Fig 6. Immune-associated pathways enriched in the downregulated genes.** This figure illustrates the major immune-related pathways that are significantly downregulated in lung cancer.

as inflammation, immune response, cell signaling, and tumor progression. Fig 8A illustrates the interaction network of the top 50 hub genes an, highlighting their interconnections and potential involvement in the molecular pathways that drive lung cancer progression. We further identified 201 common upregulated hub genes and 224 common downregulated hub genes, which were shared across the datasets. These genes represent key molecular players in the tumorigenic processes of lung cancer, with their dysregulation potentially contributing to cancer progression. The results of this analysis are presented in Fig 8B and are detailed in Supplementary Table S7 in S1 File.

The top upregulated hub genes identified in this analysis include *MMP9, CDH1, HSP90AB1, SOX2, CDKN2A, SPP1, EZH2, TIMP1, CXCL10, COL1A1, HDAC1, BRCA1,* and *CDK4*. These genes are involved in critical processes such as tumor invasion, immune modulation, cell cycle regulation, and epigenetic changes. The top downregulated hub genes identified in this analysis *include IL6, TNF, IL1B, JUN, CCL2, TLR4, FOS, PTGS2, PECAM1, PTPRC, ICAM1, FGF2, PPARG, ITGAM, CXCL12,* and *KDR*. These genes are primarily involved in immune response, inflammation, and cellular signaling pathways.

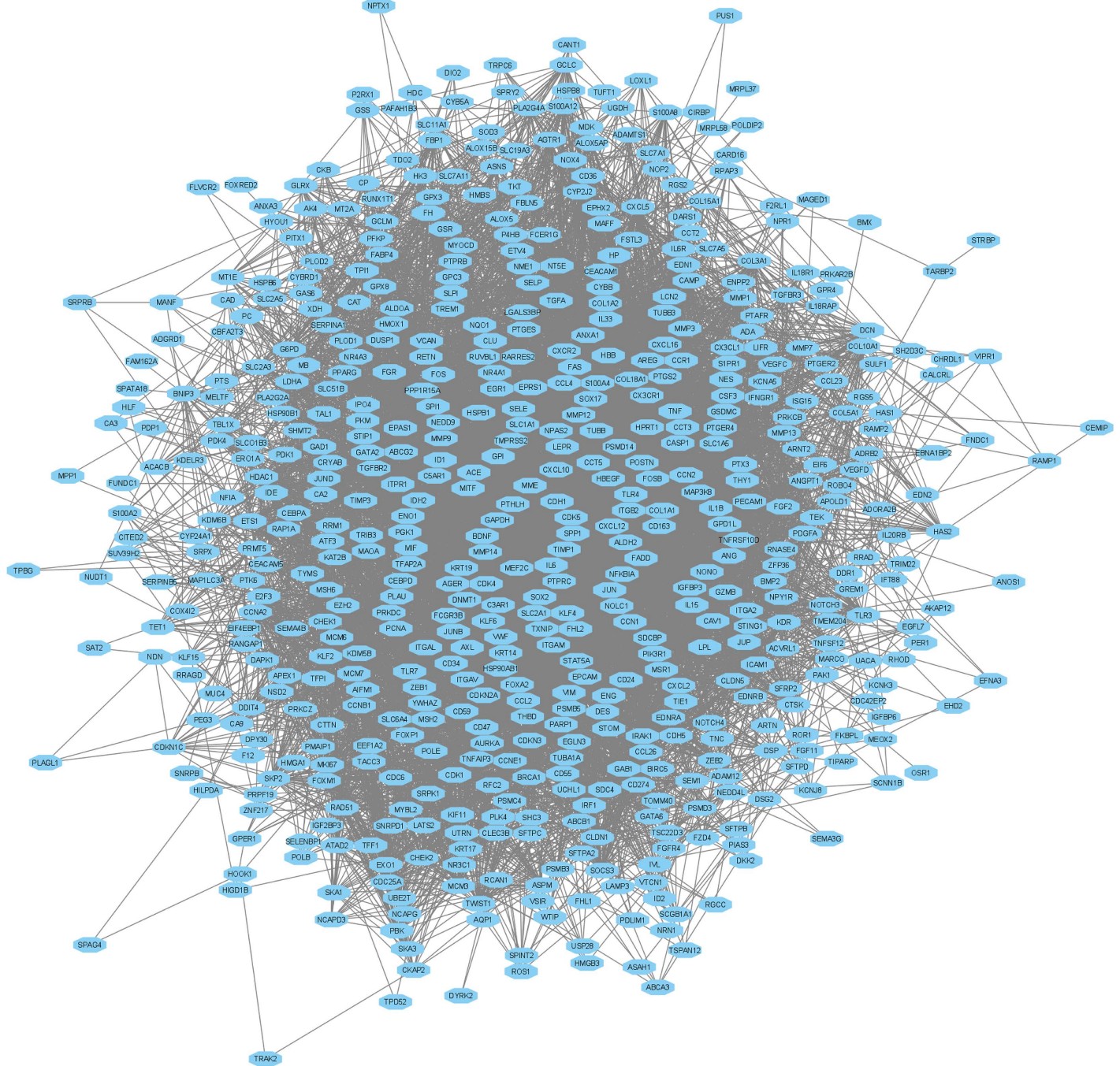

**Fig 7. Primary Protein-Protein Interaction (PPI) network of hypoxia-related differentially expressed genes (DEGs) generated using STRING.**
The network consists of 600 nodes and 9,281 edges, with an average node degree of 30.9.

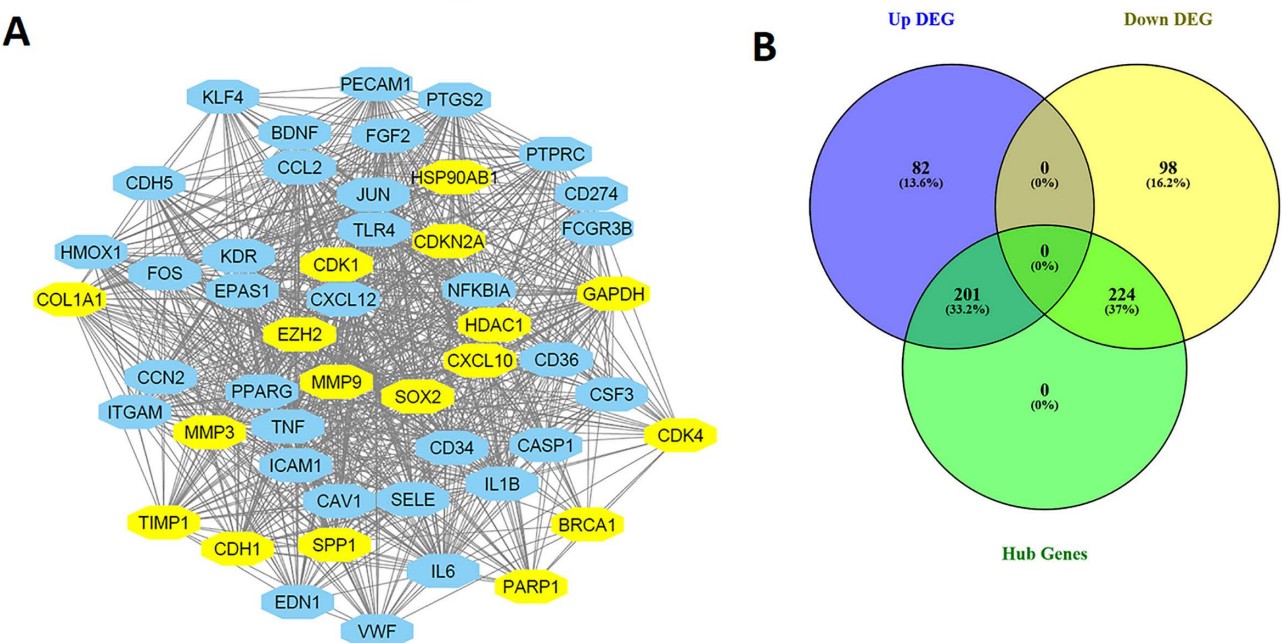

**Fig 8. Protein-Protein Interaction (PPI) Network of Hub Genes. A.** Interaction network of the top 50 hub genes identified from the PPI analysis. **B.** Common hub genes shared between the upregulated and downregulated gene sets. Yellow node represents upregulated genes and light blue node represents downregulated genes.

## Exploration of independent prognostic HRDEGs

We first conducted Kaplan-Meier survival analysis on all hub genes and identified 174 significant genes that were associated with survival (Supplementary Table S8 in S1 File). The following seven genes (Identified by Univariate regression and LASO)—*ADAM12* (HR = 1.39, P = 0.02), *CCNE1* (HR = 1.43, P = 0.01), *DSG2* (HR = 1.58, P = 0.002), *EIF6*(HR = 1.49, P = 0.008), EXO1 (HR = 1.54, P = 0.003), *IGF2 BP3* (HR = 1.6, P = 0.001), and *RAD51* (HR = 1.37, P = 0.03)—are identified as upregulated hub genes in lung cancer (Fig 9A).

The upregulated group, categorized based on median expression levels, is associated with worse survival outcomes. Increased expression of these genes is correlated with poorer prognosis, suggesting that they may play a critical role in tumor progression and could serve as potential biomarkers for identifying high-risk patients with lung cancer. The following ten genes ((Identified by Univariate regression and LASO))—*ABCB1*(HR = 0.71, P = 0.02), *ADRB2*(HR = 0.67, P = 0.008), *ALDH2*(HR = 0.66, P = 0.006), *CAT*(HR = 0.73, P = 0.03), *CLEC3B*(HR = 0.67, P = 0.006), *FBLN5*(HR = 0.64, P = 0.003), *MAP-3K8*(HR = 0.67, P = 0.006), *PIK3R1*(HR = 0.64, P = 0.003), *SFTPD*(HR = 0.68, P = 0.01), and *SOD3*(HR = 0.75, P = 0.04)—are identified as downregulated hub genes in lung cancer.

To refine this list, we performed univariate regression analysis, which revealed 106 genes with statistically significant associations (Supplementary Table S9 in S1 File). Subsequently, we applied Lasso regression to the consistently upregulated and downregulated genes to identify key genes with non-zero coefficients. This analysis led to the identification of 17 key genes: *ADRB2, ALDH2, CAT, CCNE1, MAP3K8, DSG2, EIF6, ABCB1, PIK3R1, RAD51, SFTPD, SOD3, CLEC3B, ADAM12, EXO1, FBLN5,* and *IGF2BP3*. The univariate regression analysis and LASSO regression analysis depicted in the Figs 10 and 11, respectively.

Furthermore, a multivariate Cox regression analysis was conducted to identify independent prognostic factors. The results indicated that *DSG2, EIF6,* and *EXO1* emerged as independent prognostic factors (Fig 12). The multivariate

**A**

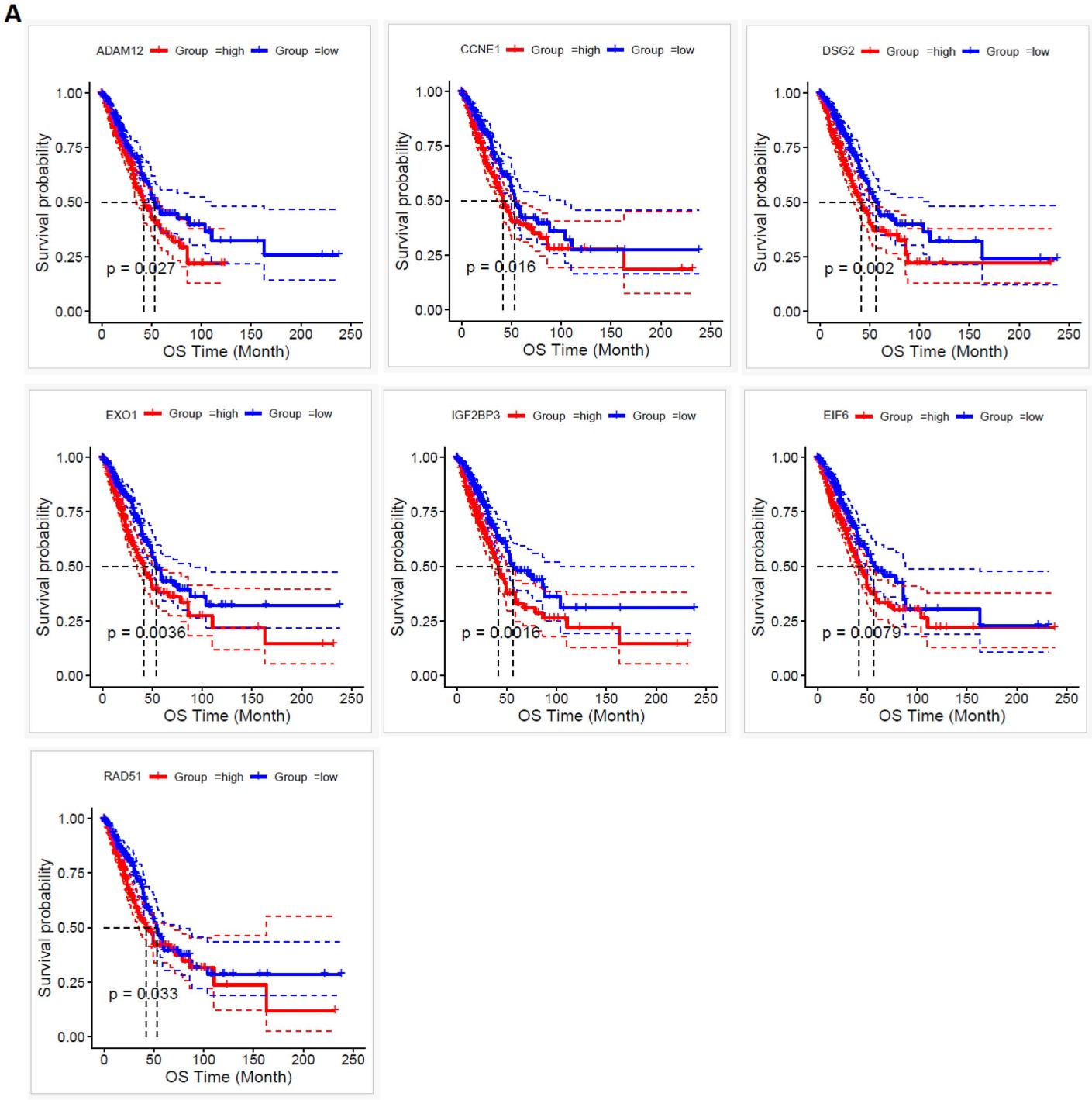

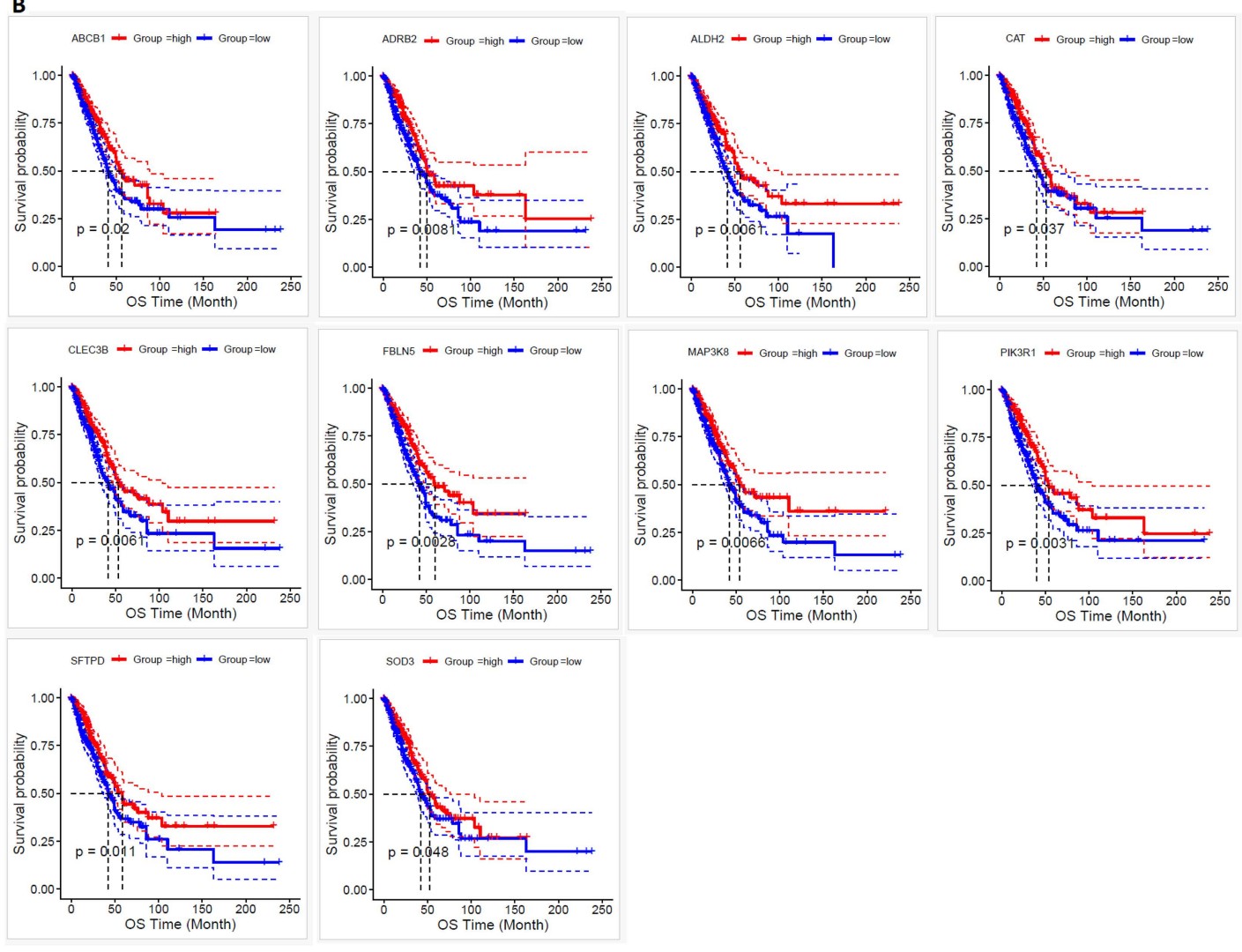

**Fig 9. Kaplan-Meier Survival Analysis of Upregulated and Downregulated Hub Genes in Lung Cancer. A.** Kaplan-Meier survival analysis of the group of upregulated hub genes, showing their significant association with survival in lung cancer patients. **(B)** Kaplan-Meier survival analysis of the group of downregulated hub genes, demonstrating their significant impact on survival outcomes.

model demonstrated a significant fit, with the following statistics: Concordance = 0.657 (SE = 0.024), Likelihood ratio test = 50.39 on 17 degrees of freedom (df), p = 4e-05, Wald test = 48.47 on 17 df (p = 7e-05), and Score (logrank) test = 48.73 on 17 df (p = 7e-05). *DSG2* (HR = 1.238, p = 0.012), *EIF6* (HR = 1.387, p = 0.046), and *EXO1* (HR = 1.257, p = 0.047) all have hazard ratios greater than 1, indicating that higher expression of these genes is associated with poorer survival outcomes.

## Validation of prognostic value in an independent cohort

To validate the prognostic relevance of *DSG2, EIF6*, and *EXO1* in our risk model, we utilized the independent data-set GSE13213 through the SurvExpress platform [33]. Cox regression model used to assess *DSG2, EIF6*, and *EXO1* included only the expression levels of these three genes. The analysis revealed that all three genes maintained a

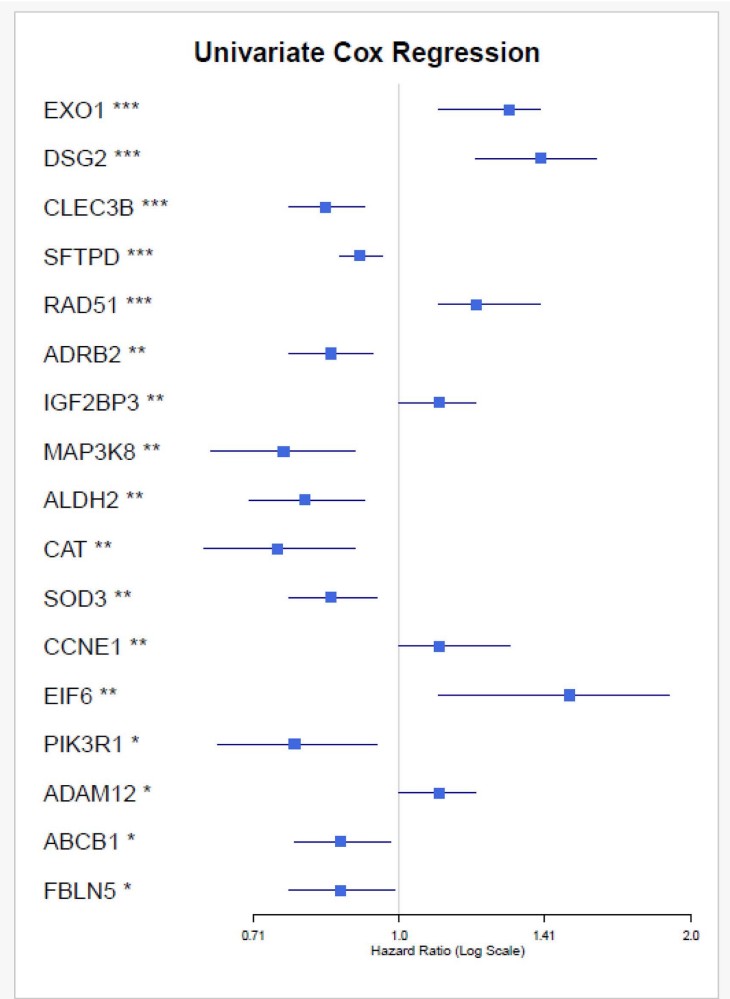

**Fig 10. The results of univariate Cox regression analysis for hub genes in lung cancer patients.** Each gene's hazard ratio (HR) with 95% confidence interval (CI) is displayed, indicating the association between gene expression and patient survival.

significant association with patient prognosis in this external cohort. Specifically, DSG2 (coefficient: 0.2795, p-value: 0.05), EIF6 (coefficient: 0.8356, p-value: 0.019), and EXO1 (coefficient: 0.7033, p-value: 0.001) were all positively correlated with poor survival outcomes. The overall model performance was statistically robust, as indicated by a Log-Rank p-value < 0.001 and a hazard ratio (HR) of 3.19, underscoring the model's strong prognostic power in stratifying patients.

Further stratification of the cohort into high-risk and low-risk groups based on the risk scores revealed a significant survival difference, with patients in the high-risk group exhibiting markedly shorter survival times compared to those in the low-risk group (Fig 13A). Moreover, expression analysis confirmed that DSG2, EIF6, and EXO1 were significantly upregulated in the high-risk group, consistent with our primary findings (Fig 13B). These validation results provide strong support for the clinical applicability of the proposed gene signature and reinforce the role of these three genes as potential biomarkers for risk stratification and prognosis in lung cancer.

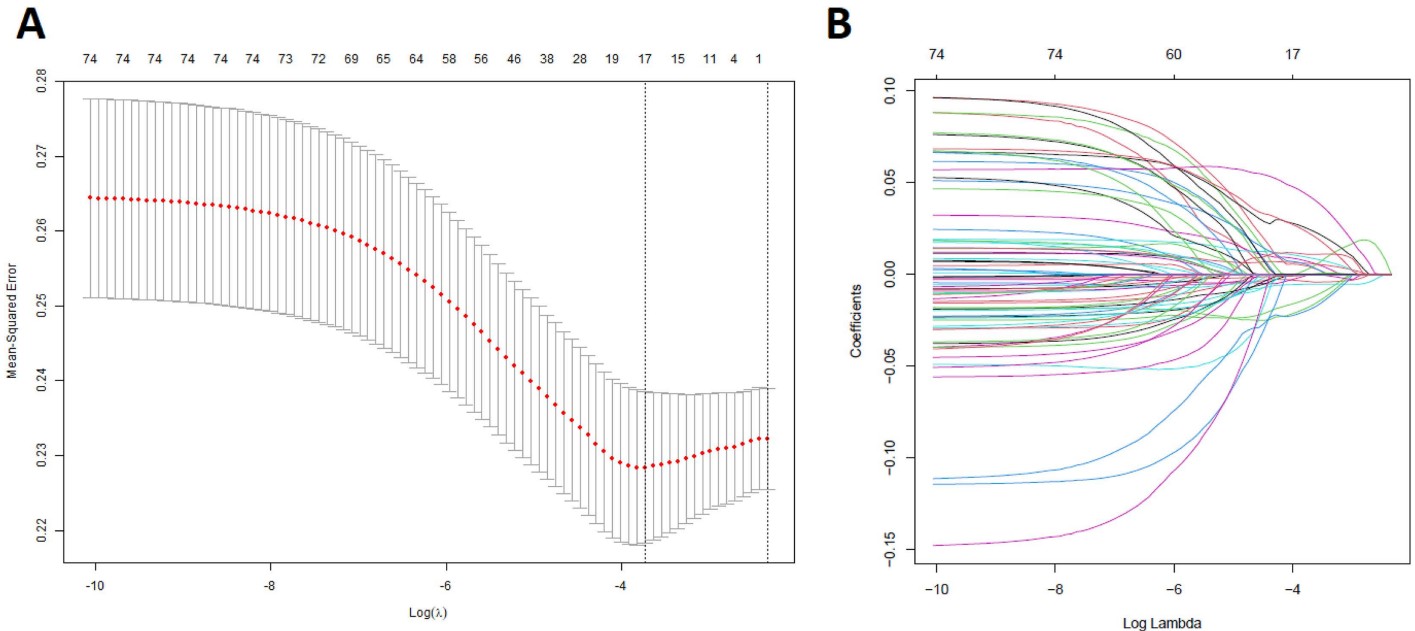

**Fig 11. Lasso regression analysis of key genes.** Identification of key genes through Lasso regression. **(A)** Lasso coefficients for each gene selected through Lasso regression. The plot shows the evolution of the coefficients for each gene as the regularization penalty (lambda) increases. **(B)** Log(lambda) vs. coefficients plot demonstrating the relationship between the log-transformed lambda values and the corresponding coefficients.

## Discussion

Lung adenocarcinoma remains one of the leading causes of cancer-related mortality worldwide, with hypoxia recognized as a key contributor to tumor progression, metastasis, and therapy resistance. In this study, we comprehensively analyzed hypoxia-related differentially expressed genes (HRDEGs) in LUAD and identified 283 upregulated and 322 downregulated HRDEGs, reflecting the broad impact of hypoxic signaling on multiple cellular processes. Functional enrichment and protein-protein interaction analyses revealed that upregulated genes were primarily involved in cancer-related and cellular signaling pathways, whereas downregulated genes were enriched in immunity-associated pathways, highlighting the dual role of hypoxia in promoting tumor growth while modulating immune responses.

The enrichment of cell cycle–related genes, including *PCNA, MCM7, CDKN2A, PRKDC, HDAC1, CDC6, CDK4, CHEK1,* and *E2F3*, highlights a pronounced dysregulation of cell cycle control mechanisms in lung cancer. These genes are central to DNA replication, cell cycle progression, and checkpoint regulation, ensuring genomic integrity under normal physiological conditions [35–37]. Their coordinated upregulation suggests enhanced proliferative capacity and impaired checkpoint surveillance, which are hallmark features of tumorigenesis. In particular, aberrant activation of CDKs and E2F transcription factors can drive uncontrolled entry into the S phase [38], while altered expression of DNA damage response genes such as *CHEK1* and *PRKDC* may allow cancer cells to bypass critical repair checkpoints, thereby promoting genomic instability [39,40]. Collectively, these findings underscore the pivotal role of cell cycle dysregulation in lung cancer development and progression.

In addition to cell cycle alterations, the HIF-1 signaling pathway (hsa04066) emerged as another significantly enriched pathway, with notable upregulation of genes including *LDHA, EGLN3, SLC2A1, EIF4EBP1, PGK1, TIMP1, ENO1, ALDOA, GAPDH, PFKP,* and *PDK1*. These genes are key mediators of cellular adaptation to hypoxic conditions and are closely associated with metabolic reprogramming in cancer. The elevated expression of glycolytic enzymes and glucose transporters, such as *SLC2A1* and *LDHA*, reflects a shift toward aerobic glycolysis (the Warburg effect), enabling tumor

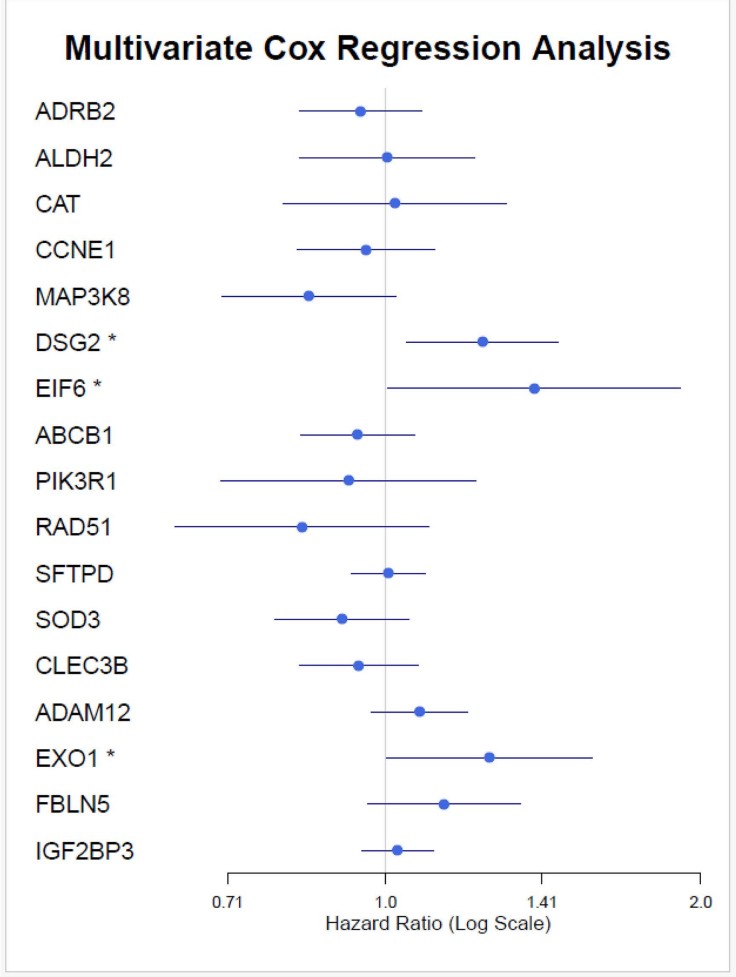

**Fig 12. Multivariate Cox regression analysis identifying independent prognostic factors.**

cells to sustain rapid growth in oxygen-deprived microenvironments [41]. Taken together, the concurrent activation of cell cycle and HIF-1 signaling pathways suggests a coordinated mechanism by which lung cancer cells promote unchecked proliferation while adapting to hypoxic stress. These pathways may act synergistically to facilitate tumor progression and aggressiveness, highlighting their potential as therapeutic targets and prognostic biomarkers in lung cancer.

Beyond the activation of oncogenic and metabolic programs, lung cancer was characterized by a marked suppression of immune-associated pathways, reflecting the establishment of an immunosuppressive tumor microenvironment [42]. The coordinated downregulation of signaling cascades involved in cytokine communication, inflammatory responses, and immune cell activation indicates a weakened immune network within tumors [43]. Furthermore, attenuation of pathways governing innate immune sensing and effector mechanisms suggests reduced recognition and clearance of malignant cells by immune surveillance systems [44]. Such widespread impairment of immune signaling is consistent with tumor-driven immune evasion, in which cancer cells actively modulate immune pathways to avoid immune-mediated destruction [45]. This immune suppression may not only facilitate tumor progression but also contribute to resistance to immune-based therapies, underscoring the critical role of immune pathway dysregulation in lung cancer pathogenesis

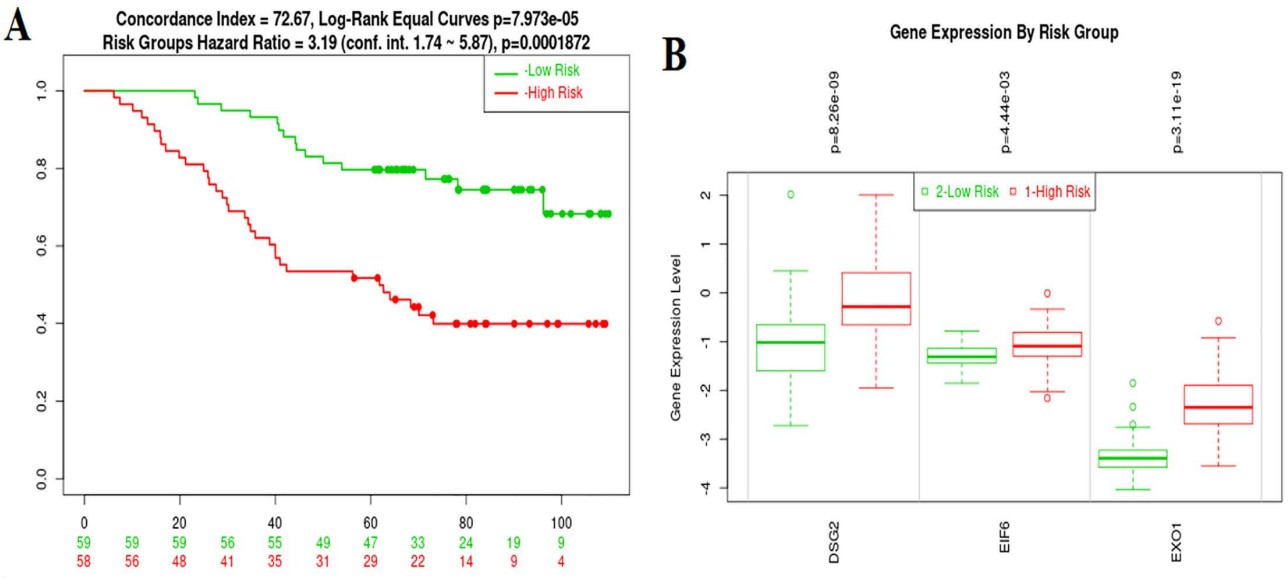

**Fig 13. Validation of prognostic gene signature using independent cohort (GSE13213).** A: Survival Analysis Based on Risk Groups in GSE13213 Dataset utilizing SurvExpress platform. B: Expression Levels of DSG2, EIF6, and EXO1 in High- and Low-Risk Groups.

[43]. Interestingly, we have previously shown that HIF-1 suppresses NF-κB signaling [46], one of the pathways found to be significantly downregulated in the analysis of lung adenocarcinoma presented here.

Moreover, the top upregulated hub genes, including, *MMP9* and *SPP1* play key roles in extracellular matrix remodeling and metastasis [47,48], while *CDH1* is important for maintaining cell-cell adhesion [49]. Other upregulated hub genes, including *SOX2* and *EZH2* regulate stem cell-like properties and epigenetic modifications, contributing to cancer stem cell maintenance and progression [50,51]. *CDK4* and *CDKN2A* are crucial regulators of the cell cycle, promoting cell division and tumor growth [52,53]. *BRCA1* and *HDAC*1 are involved in DNA repair and chromatin remodeling [54,55], while *CXCL10* and *TIMP1* are associated with immune responses [56,57]. The upregulation of these hub genes highlights their critical roles in tumor progression and their potential as therapeutic targets in lung cancer. On the other hand, the downregulation of IL6, TNF, and IL1B suggests impaired inflammatory signaling [58], while *TLR4, FOS*, and *ICAM1* are involved in immune cell activation and adhesion [59,60], indicating potential immune evasion mechanisms. *PECAM1* and *KDR* are associated with endothelial cell function and angiogenesis, both crucial for tumor vasculature [60,61]. The down-regulation of *PPARG*, *ITGAM*, and *CXCL12* further highlights disruptions in immune regulation and cell migration [62–64]. Overall, the downregulation of these hub genes suggests a weakened immune response, contributing to tumor progression and immune evasion in lung cancer.

Some of the survival associated genes are involved in the several cancers. For example, Li et al. reported that both the transcriptional and post-transcriptional expression levels of *ADAM12* are significantly elevated in tumor tissues compared to normal tissues. Furthermore, increased *ADAM12* expression was found to be associated with poor prognosis in patients with lung adenocarcinoma [65]. Ma et al. found that *CCNE1* is upregulated in LUAD and that its knockdown suppresses tumor growth and metastasis. High *CCNE1* expression correlates with advanced stage, larger tumors, lymph node involvement, and poor prognosis [66]. The downregulated group, categorized based on median expression levels, is associated with worse survival outcomes. Lower expression of these genes correlates with poorer prognosis, suggesting that their downregulation may contribute to tumor progression and aggressive cancer behavior [67,68]. These findings

highlight the potential role of these downregulated genes in worsening survival outcomes and underscore their relevance as prognostic biomarkers for lung cancer.

## Three-gene signature with prognostic value in LUAD

By integrating survival analysis and regression modeling, we further identified *DSG2, EIF6,* and *EXO1* as independent prognostic factors, establishing a novel hypoxia-related gene signature that may inform both patient prognosis and therapeutic strategies in LUAD. These genes are associated with different aspects of biology but have previously been associated with hypoxia responses [69–71]. DSG2 is ubiquitously expressed and is a type of cadherin involved in intermediate filaments, controlling cell-to-cell junctions [72]. Its deregulation is also associated with numerous diseases such as cancer and autoimmune disease [73]. *DSG2* expression is significantly associated with poor prognosis and reduced immune cell infiltration in various cancers [74]. *EIF6* encodes a translation factor that controls association of 40S and 60S, is needed for the synthesis of 60S subunits and for translation [75]. It has been shown to be overexpressed in several human cancers [74]. EXO1 is an exonuclease involved in several important biological processes such as DNA damage repair, DNA replication and meiosis [76,77]. *EXO1* has been proposed as a biomarker with both prognostic and diagnostic value across multiple cancers, highlighting its potential as a therapeutic target and its role in immunomodulation within the tumor microenvironment [77,78].

While numerous studies have reported associations between hypoxia-related genes and LUAD prognosis [18–20], our unbiased and integrative analysis identified this novel three-gene signature (*DSG2, EIF6, and EXO1*), which has not been previously reported as a prognostic panel. This signature captures distinct aspects of hypoxia-driven tumor biology: *DSG2* regulates cell adhesion and tumor progression [69], *EIF*6 controls translation initiation supporting rapid tumor growth under hypoxia [70], and *EXO1* mediates DNA repair under replication stress, which is often exacerbated in hypoxic tumors [71]. Multiple studies have demonstrated that *EXO1* is significantly overexpressed in LUAD tissues compared with adjacent normal lung tissues and that high *EXO1* expression is associated with poorer overall survival and more aggressive clinicopathological features, including advanced stage and increased tumor migration [79]. Genetic studies have demonstrated that variants in key hypoxia signaling pathways, including EXO1, are associated with poor overall survival in non–small cell lung cancer, indicating a connection between hypoxia-driven stress responses and EXO1 expression and function [80]. EIF6 has been implicated in adaptive translational control under hypoxic stress, reflecting a mechanism by which tumor cells maintain protein synthesis in low-oxygen environments, supporting survival and growth in hypoxic regions of LUAD tumors [81]. Similarly, bioinformatic analyses have identified *EIF6* overexpression in LUAD, where elevated *EIF6* correlates with enhanced tumor cell proliferation, invasion, and worse patient prognosis, suggesting its role in translational regulation and tumor progression in lung cancer [82]. Although direct studies linking *DSG2* to hypoxia are limited, *DSG2* has been reported to be upregulated in LUAD and associated with tumor progression and metastatic potential, which is consistent with its identification as an adverse prognostic factor in our study [83]. Importantly, these genes were validated in an independent cohort and may have clinical implications, as hypoxia influences the tumor microenvironment and response to immune checkpoint inhibitors. Thus, the DSG2-EIF6-EXO1 signature not only provides robust prognostic stratification but also offers mechanistic and potentially therapeutic insight, distinguishing it from previously reported hypoxia-related prognostic markers. Collectively, this evidence supports the view that *DSG2, EIF6*, and *EXO*1 are not only relevant to tumor aggressiveness in LUAD but also fit within a hypoxia-centered framework, consistent with their upregulation in more aggressive, hypoxic tumor regions and their association with poor survival in LUAD patients.

## Comparison with previous biomarker LUAD studies

Several studies have explored hypoxia-related prognostic biomarkers in LUAD, but their scope and methodology differ from our approach. Li et al. developed a 7-gene immune–hypoxia signature. Their focus was on genes at the intersection of immune and hypoxia pathways [18]. In contrast, our study analyzed a genome-wide set of hypoxia-related mRNAs, identifying three independent prognostic hub genes (DSG2, EIF6, EXO1) that do not overlap with the Li et al. signature.

Additionally, we validated our findings in independent LUAD cohorts, strengthening their clinical relevance and providing a distinct prognostic framework centered on core hypoxia-driven mRNAs. Xiong et al. focused on 15 hypoxia-related genes to classify hypoxia clusters and constructed a ceRNA-based network [19], emphasizing lncRNAs and clustering analyses. Our approach differs in analyzing a broader hypoxia-related gene set, integrating protein–protein interaction networks, and identifying independent prognostic hub genes. This allows for a more systematic understanding of hypoxia-driven mechanisms and provides stronger translational relevance for LUAD prognosis. Another research group established a 9-lncRNA hypoxia signature [20]. While informative, their study focused exclusively on non-coding RNAs. In contrast, our study emphasizes protein-coding mRNAs, identifying genes with independent prognostic value and mechanistic relevance. Our findings complement the lncRNA-based signature by highlighting additional molecular targets that can improve prognosis prediction and deepen mechanistic insights into hypoxia-driven LUAD progression. Overall, our study extends prior work by providing a comprehensive, multilayered analysis of hypoxia-related genes, integrating functional enrichment, PPI networks, and survival modeling to identify independent prognostic hub genes. This approach not only captures broader hypoxia-induced changes but also establishes a distinct prognostic model for LUAD.

## Limitations of this study

Despite the independent cohort validation and the novelty of our three-gene signature (DSG2, EIF6, and EXO1), several limitations should be acknowledged. First, the study relies on public retrospective datasets, which may introduce selection bias and limit generalizability. Second, there is cross-platform heterogeneity between datasets, including potential batch effects and annotation differences that could affect results. Third, the curated hypoxia-related gene sets used may not capture all relevant hypoxia-associated genes, potentially omitting novel or unannotated candidates. Fourth, the study lacks protein-level validation to confirm whether mRNA expression correlates with protein abundance and function. Fifth, no experimental validation using LUAD cell lines or patient-derived samples was performed, limiting mechanistic interpretation. Finally, the validation relied on an online platform (SurvExpress) rather than a fully reproduced external analytic pipeline, which may affect reproducibility.

To address these limitations, future work should include experimental validation of DSG2, EIF6, and EXO1 in LUAD models and patient samples, including qPCR or protein-level assays. Additional studies should aim to develop therapeutic strategies targeting high-risk patients identified by this signature. Moreover, efforts to harmonize datasets across platforms and expand hypoxia-related gene curation could improve the robustness and translational relevance of the findings.

## Conclusions

In conclusion, our analysis identifies *DSG2, EIF6,* and *EXO1* as key hypoxia-related genes with significant prognostic value in lung adenocarcinoma. These genes, through their involvement in hypoxia-related pathways, may play critical roles in tumor progression and patient survival. The findings suggest that *DSG2, EIF6,* and *EXO1* could serve as potential biomarkers for predicting patient outcomes and may offer novel therapeutic targets for improving prognosis in lung adenocarcinoma. Further validation of these genes in larger cohorts and clinical settings is necessary to confirm their utility in clinical practice.

## Supporting information

**S1 File. Supplementary Tables S1-S9.** S1. List of collected hypoxia related genes; Table S2. List of DEGs in TCGA LUAD; Table S3. List of DEGs in GSE18842; Table S4. List of hypoxia related DEGs in both datasets; Table S5. Enriched pathways associated with the upregulated genes (FDR < 0.05); Table S6. Enriched pathways associated with the downregulated genes (FDR < 0.05); Table S7. Rank of the PPI network by Degree method; Table S8. Significant Genes Identified in Kaplan-Meier Survival Analysis; Table S9. Significant Genes Identified in Univariate regression Analysis.
(XLSX)

## Acknowledgments

We thank all the members of the scientific community for making their datasets public.

## Author contributions

**Conceptualization:** Bandar Alghamdi.

**Data curation:** Bandar Alghamdi.

**Formal analysis:** Bandar Alghamdi.

**Funding acquisition:** Sonia Rocha.

**Methodology:** Bandar Alghamdi.

**Supervision:** Sonia Rocha.

**Visualization:** Bandar Alghamdi.

**Writing – original draft:** Bandar Alghamdi.

**Writing – review & editing:** Sonia Rocha.

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
