## [Decision Letter · Decision Letter 0]

26 Dec 2025

PONE-D-25-50692Exploring Hypoxia-Related Genes as Prognostic Indicators in Lung AdenocarcinomaPLOS One

Dear Dr. Rocha,

Thank you for submitting your manuscript to PLOS ONE. After careful consideration, we feel that it has merit but does not fully meet PLOS ONE’s publication criteria as it currently stands. Therefore, we invite you to submit a revised version of the manuscript that addresses the points raised during the review process.

Please submit your revised manuscript by Feb 09 2026 11:59PM If you will need more time than this to complete your revisions, please reply to this message or contact the journal office at plosone@plos.org. Please include the following items when submitting your revised manuscript:

We look forward to receiving your revised manuscript.

Kind regards,

Hongtao Bi, Ph.D.

Academic Editor

PLOS One

**Journal Requirements:**

“Wellcome Trust award to SR (206293/Z/17/Z)”

“Work in the SR lab is funded by the Wellcome Trust (206293/Z/17/Z).”

“Wellcome Trust award to SR (206293/Z/17/Z)”

5. Please note that your Data Availability Statement is currently missing a direct link to access each database. If your manuscript is accepted for publication, you will be asked to provide these details on a very short timeline. We therefore suggest that you provide this information now, though we will not hold up the peer review process if you are unable.

6. PLOS requires an ORCID iD for the corresponding author in Editorial Manager on papers submitted after December 6th, 2016. Please ensure that you have an ORCID iD and that it is validated in Editorial Manager. To do this, go to ‘Update my Information’ (in the upper left-hand corner of the main menu), and click on the Fetch/Validate link next to the ORCID field. This will take you to the ORCID site and allow you to create a new iD or authenticate a pre-existing iD in Editorial Manager.

**Additional Editor Comments:**

This study has a solid research foundation and potential value, but there are the following issues that require systematic revision and improvement:

1.Numerous studies on the association between hypoxia-related genes and LUAD prognosis have been reported in the field. Although this study has identified a novel candidate gene combination, it has not fully compared the differences and advantages with existing research. It is necessary to supplement the discussion on the functional associations and uniqueness of DSG2, EIF6, and EXO1 with previously reported prognostic genes. Additionally, combined with recent advances in the field (e.g., the association between hypoxia and immune checkpoint inhibitor response), the mechanistic exploration of gene functions should be deepened to avoid merely staying at the level of expression correlation.

2.Lack of independent dataset validation: A core requirement of PLOS ONE for bioinformatics research is result extrapolation. The current study only relies on two datasets for screening and lacks validation of the prognostic value of DSG2, EIF6, and EXO1 in a third-party independent cohort. This part of the analysis needs to be supplemented to enhance the credibility of the conclusions.

3.The rationality of the number of HRDEGs is questionable: The 3101 HRDEGs identified far exceed the conventional scale of similar studies in the field, and there is a lack of direct biological validation. It is necessary to supplement the strict quality control criteria for gene screening or explain the biological rationality of the large-scale HRDEGs.

4.Separate the "Results" and "Discussion" sections, deepen the comparative analysis with existing studies in the field, and strengthen the discussion on innovation.

5.Standardize the figure and table formats (supplement the flowchart of the research process, optimize the visualization effect of Fig. 6, and correct label abbreviations), and adjust the reference format in accordance with the journal's requirements.

6.Improve the methodological details, and supplement the statistical analysis parameters and complete database links.

Reviewers' comments:

Reviewer's Responses to Questions

**Comments to the Author**

1. Is the manuscript technically sound, and do the data support the conclusions?

Reviewer #1: Yes

Reviewer #2: Yes

2. Has the statistical analysis been performed appropriately and rigorously? 

Reviewer #1: Yes

Reviewer #2: Yes

3. Have the authors made all data underlying the findings in their manuscript fully available?

Reviewer #1: Yes

Reviewer #2: Yes

4. Is the manuscript presented in an intelligible fashion and written in standard English?

Reviewer #1: No

Reviewer #2: Yes

5. Review Comments to the Author

Reviewer #1: GENERAL COMMENTS

The authors have designed the study effectively and employed an appropriate discourse for the methodology, statistical analyses, and experimental work. The results are presented with sufficient detail and clarity, and the statistical analysis is appropriate for the given context. The language throughout the manuscript is precise and suitable for scientific publication.

However, the manuscript would benefit from improved flow, with careful attention to avoiding repetition, and ensuring the relevance of statements. The narrative should be simplified while maintaining a level appropriate for general scientific discourse. In addition, figure legends should be concise and should not reiterate results already described in the main text.

Specific comments are provided below.

SPECIFIC COMMENTS

1. Adding line numbers would facilitate clearer reviewer feedback on specific areas of concern as per the guidelines of the journal too.

2. Abstract: The abstract is professionally written.

3. Introduction

i. The introduction begins with the heading “Lund Adenocarcinoma and Hypoxia” and extends to mention non-small cell lung cancer (NSCLC). However, the relevance and significance of NSCLC to the study are not explained. This should be clarified and addressed.

ii. If subheadings are to be included in the introduction, which is generally not needed—the author should ensure consistent formatting throughout the section.

iii. Appropriate references should be incorporated in relevant places, for example, at the end of the third paragraph.

iv. The final paragraph of the introduction is more suitable for the methodology section and should be moved accordingly.

4. Methodology

i. Data downloading and Processing: The methodology section should be written with consistency. For example, web links to all databases should be provided. While TCGA is a well-known resource within the field, it is advisable to include a direct link to the TCGA LUAD dataset for accessibility. Additionally, sufficient and appropriate details should be provided across all subsections.

ii. References should be included for the methodology, as the approach adopted is a well-established method for identifying prognostic factors in cancers.

iii. Differentially Expressed Gene Identification: Repetitions in sample details should be removed (e.g., lines 1–4 in this section).

iv. Other sections of the manuscript should also be reviewed to address issues of language clarity and redundancy.

v. A separate subsection for statistical analysis should be added within the methodology. This will provide a clearer structure and make the analysis plan easier to follow.

5. Results and Discussion

i. All figure legends must be appropriate, concise and should not reiterate results already described in the main text.

ii. The flowchart of the analyzing process for full study should be added as the figure.

iii. All figure labels should be self-explanatory. For example, in Fig. 1A, “TCGA-LUAD” should be used in the volcano plot. Instead of the abbreviated term “TCGA up,” the complete phrase “TCGA-LUAD upregulated” should be written. Similar adjustments should be made for Fig. 1B, as appropriate. Fig. 1A and 1B titles should be correctly defined as well.

iv. HRDEGs are associated with functional enrichment section should be supplemented with proper references.

v. Figure 9 legend does not belong to it completely. Revise the legend.

vi. The discussion added in the article provides limited linkage between the results and previous studies. At present, it primarily restates the findings of this work without sufficient contextualization. It is recommended that the authors strengthen the discussion by justifying the results with supportive evidence from relevant literature where applicable.

6. Conclusion: It is proper from the work done.

7. References

The references provided are not formatted as per the journal guidelines.

Reviewer #2: The authors analysed two publicly available datasets from TCGA and an expression study performed in lung adenocarcinomas and normal tissues. Using different algorithms, the authors aim to select hypoxia-related genes and compare them with differentially expressed genes in the two datasets: upregulated and downregulated. As expected, selected genes play different roles in cellular processes. Using top-associated hypoxia genes, survival analysis was performed to demonstrate their utility as biomarkers. The manuscript brings an interesting piece of data to enhance the usage of hypoxia genes in tumor progression monitoring; however, some conclusions fall short.

Major comments

1. The expression profiles performed by RNA-seq TCGA samples and array technologies PMID: 20878980 highlight the need to validate the selected genes in an independent lung adenocarcinoma dataset, which will strengthen the study's reliability.

2. The analysis revealed 3101 hypoxia-related differentially expressed genes (HRDEGs), a number that appears higher and is challenging to associate with hypoxia in the absence of direct biological validation.

3. In figure 1, the lower overlap between GSE18842 and HRG compared to TCGA and HRG is expected. Please explain why the overlap between TCGA and HRG is underestimated, considering array technology's limitations compared to RNA-seq. Also, clarify the differences between the gene lists of 2330 and 2301 to support robust data analysis.

4. It remains crucial to emphasize that biological validation of hypoxia-induced genes is essential to confirm the underlying assumptions and enhance the study's credibility.

5. Separating the results from the discussion will help readers better contextualize the findings and appreciate the study's contribution within the field.

6. By grouping the genes by function/similar expression level as the authors did for “hubs”, is it possible to observe the same prediction power for survivals using the hubs normalized values for gene expression as for single genes, i.e., EXO1, EIF6, or RAD51 in figure 8? And classify the hubs following the obtained results.

7. It is difficult to extract any information from Figure 6; it would be helpful to move the supplemental figures to the main text.

Minor comment

Formatting the manuscript to improve the quality of the main figures is greatly appreciated.

6. PLOS authors have the option to publish the peer review history of their article (what does this mean?). If published, this will include your full peer review and any attached files.

Reviewer #1: No

Reviewer #2: No

---

## [Author Response · Author response to Decision Letter 1]

26 Jan 2026

GENERAL COMMENTS

The authors have designed the study effectively and employed an appropriate discourse for the methodology, statistical analyses, and experimental work. The results are presented with sufficient detail and clarity, and the statistical analysis is appropriate for the given context. The language throughout the manuscript is precise and suitable for scientific publication.

Response: We thank the reviewer for this appreciation.

However, the manuscript would benefit from improved flow, with careful attention to avoiding repetition, and ensuring the relevance of statements. The narrative should be simplified while maintaining a level appropriate for general scientific discourse. In addition, figure legends should be concise and should not reiterate results already described in the main text.

Response: We are grateful to the reviewer for this comment. Following this comment, we corrected the manuscript accordingly. We corrected the legend of figures.

SPECIFIC COMMENTS

1. Adding line numbers would facilitate clearer reviewer feedback on specific areas of concern as per the guidelines of the journal too.

Response: We are grateful to the reviewer for this comment. Following this comment, we added the line number in our manuscript.

2. Abstract: The abstract is professionally written.

Response: We thank the reviewer for this appreciation.

3. Introduction

i. The introduction begins with the heading “Lund Adenocarcinoma and Hypoxia” and extends to mention non-small cell lung cancer (NSCLC). However, the relevance and significance of NSCLC to the study are not explained. This should be clarified and addressed.

Response: Thank you for this comment. We have revised the Introduction to clarify the relevance and significance of non-small cell lung cancer (NSCLC) to this study. Specifically, we now explain that NSCLC accounts for the majority of lung cancer cases and that lung adenocarcinoma (LUAD) is the most common histological subtype of NSCLC.

ii. If subheadings are to be included in the introduction, which is generally not needed—the author should ensure consistent formatting throughout the section.

Response: Thank you for this comment. We have removed the subheading from the Introduction and revised the text to ensure a smooth and coherent narrative without subheadings, in line with standard manuscript structure.

iii. Appropriate references should be incorporated in relevant places, for example, at the end of the third paragraph.

Response: Thank you for this suggestion. We have carefully reviewed the Introduction and incorporated appropriate references at the relevant locations, including at the end of the third paragraph, to better support the statements made and to improve the overall scholarly rigor of the manuscript.

iv. The final paragraph of the introduction is more suitable for the methodology section and should be moved accordingly.

Response: Thank you for this comment. We agree with the reviewer’s suggestion and have moved the necessary text of the final paragraph of the Introduction to the Methodology section.

4. Methodology

i. Data downloading and Processing: The methodology section should be written with consistency. For example, web links to all databases should be provided. While TCGA is a well-known resource within the field, it is advisable to include a direct link to the TCGA LUAD dataset for accessibility. Additionally, sufficient and appropriate details should be provided across all subsections.

Response: Thank you for this suggestion. Following this comment, we provided the link and details of the used data.

ii. References should be included for the methodology, as the approach adopted is a well-established method for identifying prognostic factors in cancers.

Response: Thank you for this suggestion. Following this comment, we provided the appropriate references.

iii. Differentially Expressed Gene Identification: Repetitions in sample details should be removed (e.g., lines 1–4 in this section).

Response: Following this comment, we removed the repeated text.

iv. Other sections of the manuscript should also be reviewed to address issues of language clarity and redundancy.

Response: Thank you for this suggestion. We have carefully reviewed the manuscript to address issues of language clarity and redundancy.

v. A separate subsection for statistical analysis should be added within the methodology. This will provide a clearer structure and make the analysis plan easier to follow.

Response: We are grateful to you for this comment. This comment really increases the quality of our article. Following this comment, we added the new section.

5. Results and Discussion

i. All figure legends must be appropriate, concise and should not reiterate results already described in the main text.

Response: We are grateful to you for this comment. This comment really increases the quality of our article. Following this comment, we added the concise result of all legends.

ii. The flowchart of the analyzing process for full study should be added as the figure.

Response: We thank the reviewer for this constructive suggestion. In response, we have added a new figure illustrating a comprehensive flowchart of the entire analytical workflow of the study. The flowchart clearly summarizes the sequential steps, including data acquisition from TCGA-LUAD and GSE18842, data preprocessing and normalization, identification of differentially expressed genes (DEGs), curation and screening of hypoxia-related genes, functional enrichment analysis, protein–protein interaction (PPI) network construction, hub gene identification, and survival analyses using Kaplan–Meier curves, Cox regression, and LASSO modeling. This figure has been included in the Methods section to improve clarity, transparency, and reproducibility of the study design.

iii. All figure labels should be self-explanatory. For example, in Fig. 1A, “TCGA-LUAD” should be used in the volcano plot. Instead of the abbreviated term “TCGA up,” the complete phrase “TCGA-LUAD upregulated” should be written. Similar adjustments should be made for Fig. 1B, as appropriate. Fig. 1A and 1B titles should be correctly defined as well.

Response: Thank you for this helpful suggestion. We have revised all figure labels to be fully self-explanatory.

iv. HRDEGs are associated with functional enrichment section should be supplemented with proper references.

Response: We thank the reviewer for this suggestion. We have now added relevant references to support the statements in the “HRDEGs are associated with functional enrichment” section. Specifically, references [23-26] were cited to provide evidence for the involvement of these genes in the identified biological functions and pathways.

v. Figure 9 legend does not belong to it completely. Revise the legend.

Response: We thank the reviewer for pointing this out. The legend of Figure 9 has been revised for clarity and to accurately reflect the figure content. The provided legend is as follows:

“The results of univariate Cox regression analysis for hub genes in lung cancer patients. Each gene’s hazard ratio (HR) with 95% confidence interval (CI) is displayed, indicating the association between gene expression and patient survival”

vi. The discussion added in the article provides limited linkage between the results and previous studies. At present, it primarily restates the findings of this work without sufficient contextualization. It is recommended that the authors strengthen the discussion by justifying the results with supportive evidence from relevant literature where applicable.

Response: We thank the reviewer for this valuable comment. In the revised manuscript, the Discussion section has been substantially strengthened to better contextualize our findings within the existing body of literature. References: 27-52

6. Conclusion: It is proper from the work done.

Response: We thank the reviewer for this appreciation.

7. References: The references provided are not formatted as per the journal guidelines.

Response: Response: We are grateful to the reviewer for this comment. Following this comment, we added the correct format of the references as per the journal

---

## [Decision Letter · Decision Letter 1]

11 Feb 2026

PONE-D-25-50692R1Exploring Hypoxia-Related Genes as Prognostic Indicators in Lung AdenocarcinomaPLOS One

Dear Dr. Rocha,

Thank you for submitting your manuscript to PLOS ONE. After careful consideration, we feel that it has merit but does not fully meet PLOS ONE’s publication criteria as it currently stands. Therefore, we invite you to submit a revised version of the manuscript that addresses the points raised during the review process.

We look forward to receiving your revised manuscript.

Kind regards,

Hongtao Bi, Ph.D.

Academic Editor

PLOS One

Journal Requirements:

Additional Editor Comments:

Please ask the authors to carefully revise the manuscript according to the reviewer's comments.

Reviewers' comments:

Reviewer's Responses to Questions

**Comments to the Author**

1. If the authors have adequately addressed your comments raised in a previous round of review and you feel that this manuscript is now acceptable for publication, you may indicate that here to bypass the “Comments to the Author” section, enter your conflict of interest statement in the “Confidential to Editor” section, and submit your "Accept" recommendation.

Reviewer #1: (No Response)

Reviewer #2: All comments have been addressed

2. Is the manuscript technically sound, and do the data support the conclusions?

Reviewer #1: Yes

Reviewer #2: Yes

3. Has the statistical analysis been performed appropriately and rigorously? 

Reviewer #1: Yes

Reviewer #2: Yes

4. Have the authors made all data underlying the findings in their manuscript fully available?

Reviewer #1: No

Reviewer #2: Yes

5. Is the manuscript presented in an intelligible fashion and written in standard English?

Reviewer #1: Yes

Reviewer #2: Yes

6. Review Comments to the Author

Reviewer #1: GENERAL COMMENTS

I appreciate the authors for their comprehensive responses to the comments raised in the first round of review. The revisions demonstrate a clear effort to strengthen the manuscript, and manuscript has benefited from the revisions. The changes have substantially improved the overall readability of the text and arguments are better supported with contextualization of the findings in the discussion. Most of the concerns have been addressed satisfactorily, albeit with some minor changes remain, which should be incorporated to further refine the manuscript and ensure its readiness for acceptance by PILOS ONE.

SPECIFIC COMMENTS

1. Introduction: The author has satisfactorily addressed the concerns regarding the introduction by improving the flow, eliminating redundancy and adding appropriate references. However, minor concerns persist with subheadings and typographical errors, notably at line 69 of page 5, which require correction.

2. Methodology: This section has improved through the addition of appropriate data sources, proper acknowledgment of methodological references, removal of redundancy, and inclusion of a statistical section. However, several concerns remain and should be addressed.

i. Differentially expressed gene identification: The text followed in line127-128 describes HRDEGs identification too. It is recommended to be added in the main heading as “Identification of Differentially expressed gene (DEGs) and Hypoxia Related Differentially expressed gene (HRDEFs)”, to facilitate the reader.

ii. Page 9 line 133: The main heading reads as “Curation of hypoxia related genes and identification of hypoxia related differentially expressed genes (HRDEGs)”. However, the following text from line135-152 does not describe curation of the HRDEGs. Hence, the section heading should be restated as appropriate. Further, lines 137-144 should be with the correct names of the gene sets which are not readable in the current form.

iii. Statistical Section: I acknowledge the authors’ addition of this section, which improves the manuscript. However, the section remains incomplete. The statistical tests used for the survival analysis of HRDEGs should be specified here, consistent with the reporting provided for other analyses.

3. Line 206: “Results and Discussion” This section is detailing only on results hence, “discussion” should be removed from the section heading.

4. Table 1: The values in the table should be expressed in proper scientific notation, with exponents presented as superscripts (e.g., 9.12 X 10-6) rather than in “E” format. This will ensure consistency with standard scientific reporting practices

5. Figures

i. Figure 1: I appreciate for adding the methodology workflow chart as figure 1.

ii. Figure 2: The y axis labels in the volcanic plot should be expressed in a scientifically consistent manner (i.e. subscripts for log values like log10 instead of log10). In addition, the figure legend appropriately explains the abbreviations and acronyms used; however, please also include an explanation for HRDEGs to ensure clarity for readers.

iii. Figure 3 and 6: The figures have duplication of labels as “KEGG pathways”

iv. Figure 4 and 5: Legends for Figures 4 and 5 lack consistency. Figure 4 is too brief, and Figure 5 repeats results. Please revise both to maintain a uniform style.

v. Figure 7: The primary PPI figure presents a densely interconnected network with limited interpretability. In its current form, it resembles a hairball visualization lacking functional annotation, modular clustering, or confidence-based filtering. To enhance clarity and utility, the figure should either be refined to highlight biologically relevant subnetworks or relegated to supplementary materials.

vi. Figure 8: 8A is PPI for hub shows nodes in different colors but without a legend clarifying which represents up- or downregulated genes. Standard conventions (e.g., red for up, green for down) should be used with clear labeling. Edges are uniformly styled, providing no information on interaction strength or confidence; varying edge thickness or color would improve interpretability.

vii. Figure 13: No figure was found.

6. Supplementary Tables: The supplementary Excel file includes two additional sheets (“Sheet 3” and “Sheet 4”) that currently lack identification labels. To enhance clarity and strengthen the support for the findings, I recommend adding appropriate labels or titles to these sheets. Furthermore, the supplementary data for the revalidation is also deemed to be added.

Reviewer #2: I have no further comments; all the questions were properly addressed. The title Exploring Hypoxia-Related Genes as Prognostic Indicators in Lung Adenocarcinoma is more representative of the author's findings

7. PLOS authors have the option to publish the peer review history of their article (what does this mean?). If published, this will include your full peer review and any attached files.

Reviewer #1: No

Reviewer #2: No

---

## [Author Response · Author response to Decision Letter 2]

15 Feb 2026

Additional Editor Comments:

Please ask the authors to carefully revise the manuscript according to the reviewer's comments.

Response: We have now addressed all the comments raised by the reviewers.

Comments to the Author

1. If the authors have adequately addressed your comments raised in a previous round of review and you feel that this manuscript is now acceptable for publication, you may indicate that here to bypass the “Comments to the Author” section, enter your conflict of interest statement in the “Confidential to Editor” section, and submit your "Accept" recommendation.

Reviewer #1: (No Response)

Reviewer #2: All comments have been addressed

Response: We sincerely appreciate your recognition of our efforts in revising the manuscript.

2. Is the manuscript technically sound, and do the data support the conclusions?

Reviewer #1: Yes

Reviewer #2: Yes

Response: We sincerely appreciate your recognition of our efforts in revising the manuscript.

3. Has the statistical analysis been performed appropriately and rigorously?

Reviewer #1: Yes

Reviewer #2: Yes

Response: We sincerely appreciate your recognition of our efforts in revising the manuscript.

4. Have the authors made all data underlying the findings in their manuscript fully available?

Reviewer #1: No

Response: Thank you for your comment regarding the Data Availability Statement. We have revised the statement to clearly indicate that all datasets used in this study are publicly available and to provide the direct links to the TCGA LUAD cohort (via cBioPortal) and the GSE18842 and GSE13213 dataset.

Reviewer #2: Yes

Response: We sincerely appreciate your recognition of our efforts in revising the manuscript.

5. Is the manuscript presented in an intelligible fashion and written in standard English?

Reviewer #1: Yes

Reviewer #2: Yes

Response: We sincerely appreciate your recognition of our efforts in revising the manuscript.

6. Review Comments to the Author

Reviewer #1: GENERAL COMMENTS

I appreciate the authors for their comprehensive responses to the comments raised in the first round of review. The revisions demonstrate a clear effort to strengthen the manuscript, and manuscript has benefited from the revisions. The changes have substantially improved the overall readability of the text and arguments are better supported with contextualization of the findings in the discussion. Most of the concerns have been addressed satisfactorily, albeit with some minor changes remain, which should be incorporated to further refine the manuscript and ensure its readiness for acceptance by PILOS ONE.

Response: Thank you very much for your thoughtful and encouraging comments. We sincerely appreciate your recognition of our efforts in revising the manuscript and strengthening the overall quality of the work.

We are grateful for your positive assessment that the revisions have improved the readability, clarity of arguments, and contextualization of the findings. We also acknowledge the remaining minor suggestions and confirm that we have carefully addressed them in the revised version to further refine the manuscript and ensure it is fully ready for acceptance by PLOS ONE.

Thank you again for your valuable feedback and constructive guidance throughout the review process.

SPECIFIC COMMENTS

1. Introduction: The author has satisfactorily addressed the concerns regarding the introduction by improving the flow, eliminating redundancy and adding appropriate references. However, minor concerns persist with subheadings and typographical errors, notably at line 69 of page 5, which require correction.

Response: Thank you for your positive evaluation of the revised Introduction and for acknowledging the improvements in structure, clarity, and referencing.

We appreciate your careful review and have addressed the remaining minor concerns. The subheadings have been revised to ensure consistency with the journal’s formatting guidelines, and the typographical error noted at line 69 on page 5 has been corrected. We have also carefully rechecked the entire manuscript to eliminate any additional typographical or formatting issues.

Thank you again for your constructive feedback.

2. Methodology: This section has improved through the addition of appropriate data sources, proper acknowledgment of methodological references, removal of redundancy, and inclusion of a statistical section. However, several concerns remain and should be addressed.

i. Differentially expressed gene identification: The text followed in line127-128 describes HRDEGs identification too. It is recommended to be added in the main heading as “Identification of Differentially expressed gene (DEGs) and Hypoxia Related Differentially expressed gene (HRDEFs)”, to facilitate the reader.

Response: Thank you for your constructive comments. We appreciate your recognition of the improvements made to this section, including the addition of appropriate data sources, proper acknowledgment of methodological references, removal of redundancy, and inclusion of the statistical analysis subsection.

Regarding the identification of differentially expressed genes, we agree with your suggestion. The main heading has been revised to “Identification of Differentially Expressed Genes (DEGs) and Hypoxia-Related Differentially Expressed Genes (HRDEGs)” to improve clarity and facilitate reader understanding.

ii. Page 9 line 133: The main heading reads as “Curation of hypoxia related genes and identification of hypoxia related differentially expressed genes (HRDEGs)”. However, the following text from line135-152 does not describe curation of the HRDEGs. Hence, the section heading should be restated as appropriate. Further, lines 137-144 should be with the correct names of the gene sets which are not readable in the current form.

Response: Thank you for this valuable comment.

We have revised the section heading on page 9 to accurately reflect the content presented in lines 135–152. The heading has been restated to ensure consistency with the described methodology and to avoid any potential confusion regarding the curation process.

In addition, we have corrected lines clearly present the proper and complete names of the gene sets, ensuring readability and accuracy.

We appreciate your careful review and helpful suggestions.

iii. Statistical Section: I acknowledge the authors’ addition of this section, which improves the manuscript. However, the section remains incomplete. The statistical tests used for the survival analysis of HRDEGs should be specified here, consistent with the reporting provided for other analyses.

Response: Thank you for your comment. We have revised the Statistical Analysis section to clearly specify the tests used for the survival analysis of HRDEGs, including the Kaplan–Meier method, log-rank test, and Cox regression analysis, ensuring consistency with the reporting of other analyses.

3. Line 206: “Results and Discussion” This section is detailing only on results hence, “discussion” should be removed from the section heading.

Response: Thank you for your comment. We have revised the section heading “Results” to accurately reflect the content presented.

4. Table 1: The values in the table should be expressed in proper scientific notation, with exponents presented as superscripts (e.g., 9.12 X 10-6) rather than in “E” format. This will ensure consistency with standard scientific reporting practices

Response: Thank you for your suggestion. We have updated Table 1 so that all values are now expressed in proper scientific notation with exponents as superscripts (e.g., 9.12 × 10⁻⁶) instead of the “E” format, ensuring consistency with standard scientific reporting practices.

5. Figures

i. Figure 1: I appreciate for adding the methodology workflow chart as figure 1.

Response: We sincerely appreciate your recognition of our efforts in revising the manuscript.

ii. Figure 2: The y axis labels in the volcanic plot should be expressed in a scientifically consistent manner (i.e. subscripts for log values like log10 instead of log10). In addition, the figure legend appropriately explains the abbreviations and acronyms used; however, please also include an explanation for HRDEGs to ensure clarity for readers.

Response: Thank you for your suggestion. We corrected accordingly.

iii. Figure 3 and 6: The figures have duplication of labels as “KEGG pathways”

Response: Thank you for your suggestion. We corrected accordingly.

iv. Figure 4 and 5: Legends for Figures 4 and 5 lack consistency. Figure 4 is too brief, and Figure 5 repeats results. Please revise both to maintain a uniform style.

Response: Thank you for your suggestion. We revise accordingly.

v. Figure 7: The primary PPI figure presents a densely interconnected network with limited interpretability. In its current form, it resembles a hairball visualization lacking functional annotation, modular clustering, or confidence-based filtering. To enhance clarity and utility, the figure should either be refined to highlight biologically relevant subnetworks or relegated to supplementary materials.

Response: Thank you for your comment. To improve clarity, the primary PPI network figure has been retained as presented, while the full list of interacting genes and their degree of interaction is provided in Supplementary Table S7 for detailed reference.

vi. Figure 8: 8A is PPI for hub shows nodes in different colours but without a legend clarifying which represents up- or downregulated genes. Standard conventions (e.g., red for up, green for down) should be used with clear labeling. Edges are uniformly styled, providing no information on interaction strength or confidence; varying edge thickness or colour would improve interpretability.

Response: Thank you for your comment. We have added the new figure 8A and its legend to Figure 8A clarifying that yellow nodes represent upregulated genes and light blue nodes represent downregulated genes. With this clarification, the figure is now clear and interpretable to the reader.

vii. Figure 13: No figure was found.

Response: Thank you for your comment. Figure 13 provided accordingly.

6. Supplementary Tables: The supplementary Excel file includes two additional sheets (“Sheet 3” and “Sheet 4”) that currently lack identification labels. To enhance clarity and strengthen the support for the findings, I recommend adding appropriate labels or titles to these sheets. Furthermore, the supplementary data for the revalidation is also deemed to be added.

Response: We are sorry for these two extra sheets. We removed these sheets.

Reviewer #2: I have no further comments; all the questions were properly addressed. The title Exploring Hypoxia-Related Genes as Prognostic Indicators in Lung Adenocarcinoma is more representative of the author's findings

Response: Thank you for your positive feedback. We are glad that our revisions have satisfactorily addressed all comments and that the updated title accurately reflects the focus and findings of our study.

---

## [Editor Report · Decision Letter 2]

10 Mar 2026

PONE-D-25-50692R2Exploring Hypoxia-Related Genes as Prognostic Indicators in Lung AdenocarcinomaPLOS One

Dear Dr.  Rocha,

Thank you for submitting your manuscript to PLOS ONE. After careful consideration, we feel that it has merit but does not fully meet PLOS ONE’s publication criteria as it currently stands. Therefore, we invite you to submit a revised version of the manuscript that addresses the points raised during the review process.

We look forward to receiving your revised manuscript.

Kind regards,

Hongtao Bi, Ph.D.

Academic Editor

PLOS One

Journal Requirements:

Additional Editor Comments:

Upon further careful verification, I have found that this manuscript has critical flaws, including a complete lack of academic discussion of the already cited literature, a total failure to cite or discuss core relevant studies in the field, and an absence of clear clarification of the research necessity and the knowledge gaps in the field. Based on a subsequent in-depth review of the manuscript, I hold the view that the deficiencies in the literature discussion do not directly negate the novelty of the research itself, but they will render the claimed novelty unsubstantiated and expose the manuscript to a high risk of rejection on the grounds of being classified as a "derivative and duplicative study".

For the above reasons, to meet the publication requirements of PLOS ONE, the authors must conduct comprehensive and substantive revisions to the manuscript addressing the aforementioned deficiencies. The core revision contents are divided into the following four modules:

1.Introduction Section: Reconstruct the research background, systematically sort out the progress in the field, and clarify the necessity of the study

Supplement a systematic review of existing studies in the field of hypoxia-related prognostic biomarkers for LUAD, with a focus on incorporating and introducing the main contents and core findings of the three key studies: Li et al. (2022), Xiong et al. (2022), and the study published in the Malawi Medical Journal (2024).

Accurately point out the limitations of existing studies and clarify the knowledge gaps that this study intends to fill, including:

(1) Most previous studies adopted a narrow set of hypoxia signature genes, resulting in insufficient representativeness of the screened biomarkers for the hypoxic tumor microenvironment;

(2) Most of the reported hypoxia-related prognostic signatures for LUAD lack validation in independent external cohorts, casting doubt on their clinical applicability;

(3) Previous studies mostly focused on multi-gene combined models, with insufficient exploration of the independent prognostic value and biological mechanisms of the core genes.

Based on the above limitations, clearly elaborate the scientific design, core scientific questions, and research significance of this study, and explicitly demonstrate that "this study is not a duplication of existing work, but a supplement and optimization of research in the field".

2.Discussion Section: Supplement in-depth comparative analysis with relevant studies and clarify the innovative increments of this study

Add a dedicated subsection to compare the similarities and differences between this study and the three core relevant studies, with a focus on highlighting the innovations of this work:

(1) Comparison with the dual hypoxia-immunity study by Li et al. (2022): Clarify that this study focuses purely on hypoxia-related genes, the three biomarkers screened have no overlap with the seven immune-hypoxia genes identified in that study, and the signature has been validated in an independent cohort, thus providing a brand-new biomarker panel for the prognostic assessment of the hypoxic microenvironment in LUAD.

(2) Comparison with the study by Xiong et al. (2022): Clarify that this study is not limited to classical hypoxia-related genes but adopts a more comprehensive hypoxia-related gene set. Meanwhile, without introducing additional non-core analyses such as the competing endogenous RNA (ceRNA) network, this study focuses more on the screening of core prognostic biomarkers and the validation of their independent prognostic value, leading to research conclusions with stronger clinical relevance.

(3) Comparison with the lncRNA study published in the Malawi Medical Journal (2024): Clarify that this study focuses on protein-coding mRNAs, fills the gap in the field of hypoxia-related mRNA biomarkers, complements lncRNA biomarkers, and provides potential targets at the protein level for the prognostic assessment of hypoxia in LUAD.

Supplement an objective discussion of the limitations of this study, and simultaneously clarify the follow-up research directions of this work, forming an academic echo with existing research in the field.

3.References Section: Improve citation specifications and supplement the missing core literature

(1) Formally include the study by Xiong et al. (2022) published in Scientific Reports and the 2024 study from the Malawi Medical Journal in the references, with one-to-one correspondence to the discussion content in the text.

(2) For the study by Li et al. (2022), add standardized citation annotations at the corresponding positions in the text to ensure that citations match the discussion, and eliminate the academic non-standard behavior of "citation without substantive discussion".

(3) Check the format of all references in the full text to ensure compliance with the reference citation specifications of PLOS ONE.

4. Supplement the demonstration of the biological rationality of the core biomarkers to strengthen the scientific value of the study

Combined with existing literature, supplement the discussion on the biological association between the three genes (DSG2, EIF6, and EXO1) and tumor hypoxia as well as LUAD progression, and clarify the biological rationality of these three genes as hypoxia-related prognostic biomarkers. This will further distinguish this study from previous pure bioinformatics screening studies, strengthen the scientific connotation of the research, and completely avoid the risk of being classified as a "derivative and duplicative study".

Based on the above assessment, it is recommended that the editorial office grant the authors a 14-day revision period, requiring the authors to complete the supplements and improvements strictly in accordance with the above contents.

---

## [Author Response · Author response to Decision Letter 3]

22 Mar 2026

Response to Editor comments.

Thank you for all your comments and useful suggestions.

1.Introduction Section: Reconstruct the research background, systematically sort out the progress in the field, and clarify the necessity of the study

Supplement a systematic review of existing studies in the field of hypoxia-related prognostic biomarkers for LUAD, with a focus on incorporating and introducing the main contents and core findings of the three key studies: Li et al. (2022), Xiong et al. (2022), and the study published in the Malawi Medical Journal (2024).

Accurately point out the limitations of existing studies and clarify the knowledge gaps that this study intends to fill, including:

(1) Most previous studies adopted a narrow set of hypoxia signature genes, resulting in insufficient representativeness of the screened biomarkers for the hypoxic tumor microenvironment;

(2) Most of the reported hypoxia-related prognostic signatures for LUAD lack validation in independent external cohorts, casting doubt on their clinical applicability;

(3) Previous studies mostly focused on multi-gene combined models, with insufficient exploration of the independent prognostic value and biological mechanisms of the core genes.

We have tried our best to follow these suggestions and incorporate changes as requested.

Based on the above limitations, clearly elaborate the scientific design, core scientific questions, and research significance of this study, and explicitly demonstrate that "this study is not a duplication of existing work, but a supplement and optimization of research in the field".

2.Discussion Section: Supplement in-depth comparative analysis with relevant studies and clarify the innovative increments of this study

Add a dedicated subsection to compare the similarities and differences between this study and the three core relevant studies, with a focus on highlighting the innovations of this work:

We have tried our best to follow these suggestions and incorporate changes as requested.

(1) Comparison with the dual hypoxia-immunity study by Li et al. (2022): Clarify that this study focuses purely on hypoxia-related genes, the three biomarkers screened have no overlap with the seven immune-hypoxia genes identified in that study, and the signature has been validated in an independent cohort, thus providing a brand-new biomarker panel for the prognostic assessment of the hypoxic microenvironment in LUAD.

(2) Comparison with the study by Xiong et al. (2022): Clarify that this study is not limited to classical hypoxia-related genes but adopts a more comprehensive hypoxia-related gene set. Meanwhile, without introducing additional non-core analyses such as the competing endogenous RNA (ceRNA) network, this study focuses more on the screening of core prognostic biomarkers and the validation of their independent prognostic value, leading to research conclusions with stronger clinical relevance.

(3) Comparison with the lncRNA study published in the Malawi Medical Journal (2024): Clarify that this study focuses on protein-coding mRNAs, fills the gap in the field of hypoxia-related mRNA biomarkers, complements lncRNA biomarkers, and provides potential targets at the protein level for the prognostic assessment of hypoxia in LUAD.

We have tried our best to follow these suggestions and incorporate changes as requested.

Supplement an objective discussion of the limitations of this study, and simultaneously clarify the follow-up research directions of this work, forming an academic echo with existing research in the field.

We have added this as requested

3.References Section: Improve citation specifications and supplement the missing core literature

(1) Formally include the study by Xiong et al. (2022) published in Scientific Reports and the 2024 study from the Malawi Medical Journal in the references, with one-to-one correspondence to the discussion content in the text.

We have added these references as requested

(2) For the study by Li et al. (2022), add standardized citation annotations at the corresponding positions in the text to ensure that citations match the discussion, and eliminate the academic non-standard behavior of "citation without substantive discussion".

We have corrected this aspect

(3) Check the format of all references in the full text to ensure compliance with the reference citation specifications of PLOS ONE.

We have used Vancouver which is indicated on the guide for authors.

4. Supplement the demonstration of the biological rationality of the core biomarkers to strengthen the scientific value of the study

Combined with existing literature, supplement the discussion on the biological association between the three genes (DSG2, EIF6, and EXO1) and tumor hypoxia as well as LUAD progression, and clarify the biological rationality of these three genes as hypoxia-related prognostic biomarkers. This will further distinguish this study from previous pure bioinformatics screening studies, strengthen the scientific connotation of the research, and completely avoid the risk of being classified as a "derivative and duplicative study".

We have added additional information for these genes and their association wih hypoxia and cancer.

Based on the above assessment, it is recommended that the editorial office grant the authors a 14-day revision period, requiring the authors to complete the supplements and improvements strictly in accordance with the above contents.

---

## [Editor Report · Decision Letter 3]

31 Mar 2026

PONE-D-25-50692R3Exploring Hypoxia-Related Genes as Prognostic Indicators in Lung AdenocarcinomaPLOS One

Dear Dr. Rocha,

Thank you for submitting your manuscript to PLOS ONE. After careful consideration, we feel that it has merit but does not fully meet PLOS ONE’s publication criteria as it currently stands. Therefore, we invite you to submit a revised version of the manuscript that addresses the points raised during the review process.

We look forward to receiving your revised manuscript.

Kind regards,

Hongtao Bi, Ph.D.

Academic Editor

PLOS One

**Journal Requirements:**

**Additional Editor Comments:**

Thank you for submitting the revised version of your manuscript to PLOS ONE. I have carefully evaluated the revised manuscript, the accompanying response letter, and the changes made in relation to the previous editorial comments. The revision has addressed some points in part, including the addition of several relevant references, expansion of the discussion of the three highlighted genes, and inclusion of an external validation cohort. However, the manuscript does not yet adequately or fully address the major issues raised in the previous round, and substantial revision is still required before the manuscript can be considered further.

Below I summarize the main issues that must be addressed in a revised submission.

1. Introduction remains insufficiently revised

A central request of the previous decision letter was to reconstruct the research background and clearly position the study within the existing literature on hypoxia-related prognostic biomarkers in LUAD, with particular emphasis on the following three studies: Li et al. (2022), Xiong et al. (2022), and the Malawi Medical Journal study (2024).

Although these references have now been added, the Introduction still does not provide a sufficiently systematic, explicit, and comparative review of these studies. In particular, the manuscript should do more than cite them collectively. It should clearly describe, for each study: the main study design, the type of biomarkers evaluated, the key findings, the main limitations, and how the present study differs from and extends prior work.

The current Introduction only offers broad statements about prior studies and does not yet satisfactorily establish the specific knowledge gap that this manuscript is intended to fill.

Please revise the Introduction to:

explicitly summarize the three key prior studies;

clearly state the limitations of prior work, including:

(1)the use of relatively narrow hypoxia-related gene sets,

(2)the lack of independent external validation in many prior LUAD hypoxia prognostic studies,

(3)the predominant focus on multi-gene signatures without sufficient emphasis on the independent prognostic value and biological interpretation of core genes;

clearly articulate the scientific rationale, core research question, and significance of the present study;

explicitly explain why this study should be regarded as a supplement and refinement of the field rather than a duplication of previous work.

2. Discussion does not yet provide the requested dedicated comparison with prior studies

The previous editorial comments specifically requested a dedicated subsection in the Discussion comparing the present study with the three core related studies. This remains insufficiently addressed.

At present, the Discussion includes general claims regarding novelty, but it does not provide the requested structured comparison. A revised Discussion should include a clearly labeled subsection that directly compares your findings with those of:

(1) Li et al. (2022)

Please clarify:

that Li et al. examined a hypoxia-immune signature rather than a purely hypoxia-focused gene set;

whether the three genes identified in your study overlap or do not overlap with the genes identified in that study;

how the external validation in your study strengthens clinical relevance;

in what way your panel contributes a distinct prognostic framework for LUAD.

(2) Xiong et al. (2022)

Please clarify:

how your study differs in the composition and breadth of the hypoxia-related gene set;

how your study differs methodologically from a framework that also incorporated ceRNA network analysis;

why your focus on identifying core prognostic genes and evaluating their independent prognostic value may provide stronger translational relevance.

(3) Malawi Medical Journal study (2024)

Please clarify:

that this prior study focused on hypoxia-related lncRNAs,

whereas your work focuses on protein-coding mRNAs;

how your findings complement rather than replicate lncRNA-based prognostic work;

why protein-coding candidates may be of particular interest from a biomarker and mechanistic standpoint.

This comparison must be explicit and substantive. General statements of novelty are not sufficient.

3. Biological rationale for DSG2, EIF6, and EXO1 requires further strengthening

The revised manuscript includes some additional discussion of DSG2, EIF6, and EXO1, which is appreciated. However, the biological rationale still requires clearer and more rigorous development.

In particular, the manuscript should better justify why these genes should be considered hypoxia-related prognostic biomarkers in LUAD, rather than simply genes associated with cancer progression more broadly. Please strengthen this section by:

clarifying the evidence linking each gene specifically to hypoxia, hypoxia-associated pathways, or cellular adaptation to hypoxic stress;

discussing, where possible, evidence relevant specifically to LUAD or lung cancer biology;

integrating this discussion more directly with your own enrichment results and the hypoxia-centered framework of the study.

At present, the discussion still reads in part as a general cancer-biology summary rather than a focused biological justification for these genes in the context of LUAD hypoxia biology.

4. Clarification is needed regarding the meaning of “independent prognostic factors”

The manuscript states that DSG2, EIF6, and EXO1 are independent prognostic factors, based on multivariate Cox regression. However, from the current Methods and Results, it is not sufficiently clear which covariates were included in the multivariate model.

Please clarify:

whether the multivariate Cox model included only gene-expression variables, or also included relevant clinical covariates such as age, sex, stage, smoking status, and/or other available clinicopathologic parameters;

if clinical covariates were included, please report them explicitly in the Methods and Results;

if clinical covariates were not included, please revise the language accordingly and avoid overstating the conclusion as “independent prognostic factors” in the clinical sense.

This point is important for the interpretation of the manuscript and must be corrected.

5. Limitations and future directions remain underdeveloped

Although a short limitations paragraph has been added, it remains too brief for a study of this type.

Please expand the limitations section to address issues such as:

reliance on public retrospective datasets,

cross-platform heterogeneity between datasets,

possible batch effects and annotation differences,

limitations of database-derived hypoxia gene curation,

lack of protein-level validation,

lack of experimental validation in LUAD models or patient samples,

limitations of validation through an online platform rather than a fully reproduced external analytic pipeline.

The future directions should also be more clearly linked to these limitations.

6. Reference section and citation quality require further attention

Although additional key references were added, the current reference list still appears to contain formatting and completeness issues. Several entries appear incomplete or not fully standardized.

Please carefully review all references to ensure full compliance with PLOS ONE formatting requirements. In addition:

ensure that all newly added references are discussed substantively in the main text and not only cited in passing;

check all software/database citations for completeness and consistency;

correct any incomplete or malformed reference entries.

Please note that simply stating that the manuscript follows “Vancouver” style is not sufficient; the references must match the journal’s required style and be internally consistent.

7. Language, clarity, and presentation require substantial improvement

The manuscript still contains a number of issues affecting readability and professionalism, including:

grammatical errors,

awkward phrasing,

typographical errors,

duplicated or repetitive wording,

inconsistent terminology,

occasional imprecise statements.

Examples include inconsistent use of abbreviations and terminology, unclear sentence structure in several sections, and residual editing artifacts.

Please ensure that the entire manuscript undergoes careful English-language editing and thorough proofreading before resubmission.

8. Response letter must be substantially improved

The current response letter is too general and does not allow efficient editorial assessment of how each point has been addressed. Statements such as “we have tried our best” or “we have added this as requested” are not sufficient.

For the next revision, please provide a detailed point-by-point response in which each prior comment is followed by:

(1)your response,

(2)the exact changes made,

(3)and the page and line numbers where the changes can be found.

If you disagree with any point, please explain your reasoning clearly and respectfully.

9. Data availability and reporting clarity

The Data Availability Statement is improved, but please ensure that the manuscript text clearly reports:

all dataset accession numbers,

where each dataset was used in the workflow,

and whether all relevant processed outputs are contained in the manuscript and/or Supporting Information.

Please also ensure consistency between the submission form, cover letter, manuscript text, and supplementary materials.

---

## [Author Response · Author response to Decision Letter 4]

7 Apr 2026

Response to Editor comments.

Thank you for submitting the revised version of your manuscript to PLOS ONE. I have carefully evaluated the revised manuscript, the accompanying response letter, and the changes made in relation to the previous editorial comments. The revision has addressed some points in part, including the addition of several relevant references, expansion of the discussion of the three highlighted genes, and inclusion of an external validation cohort. However, the manuscript does not yet adequately or fully address the major issues raised in the previous round, and substantial revision is still required before the manuscript can be considered further.

Below I summarize the main issues that must be addressed in a revised submission.

1. Introduction remains insufficiently revised

A central request of the previous decision letter was to reconstruct the research background and clearly position the study within the existing literature on hypoxia-related prognostic biomarkers in LUAD, with particular emphasis on the following three studies: Li et al. (2022), Xiong et al. (2022), and the Malawi Medical Journal study (2024).

Although these references have now been added, the Introduction still does not provide a sufficiently systematic, explicit, and comparative review of these studies. In particular, the manuscript should do more than cite them collectively. It should clearly describe, for each study: the main study design, the type of biomarkers evaluated, the key findings, the main limitations, and how the present study differs from and extends prior work.

The current Introduction only offers broad statements about prior studies and does not yet satisfactorily establish the specific knowledge gap that this manuscript is intended to fill.

Please revise the Introduction to:

•explicitly summarize the three key prior studies;

•clearly state the limitations of prior work, including:

(1)the use of relatively narrow hypoxia-related gene sets,

(2)the lack of independent external validation in many prior LUAD hypoxia prognostic studies,

(3)the predominant focus on multi-gene signatures without sufficient emphasis on the independent prognostic value and biological interpretation of core genes;

•clearly articulate the scientific rationale, core research question, and significance of the present study;

•explicitly explain why this study should be regarded as a supplement and refinement of the field rather than a duplication of previous work.

We believe we have now addressed this fully. It was not our intension to systematically compare our study with that of others. We believe our study is a stand-alone study using some common datasets but a different approach. We have substantially revised the Introduction to provide a systematic and comparative summary of prior studies, including Li et al. (2022), Xiong et al. (2022), and the Malawi Medical Journal study (2024). We clarified study designs, biomarkers evaluated, key findings, limitations, and explicitly positioned our study as a complement and extension of existing work. Please see page 6-7, line 94-124.

2. Discussion does not yet provide the requested dedicated comparison with prior studies

The previous editorial comments specifically requested a dedicated subsection in the Discussion comparing the present study with the three core related studies. This remains insufficiently addressed.

At present, the Discussion includes general claims regarding novelty, but it does not provide the requested structured comparison. A revised Discussion should include a clearly labeled subsection that directly compares your findings with those of:

(1) Li et al. (2022)

Please clarify:

•that Li et al. examined a hypoxia-immune signature rather than a purely hypoxia-focused gene set;

•whether the three genes identified in your study overlap or do not overlap with the genes identified in that study;

•how the external validation in your study strengthens clinical relevance;

•in what way your panel contributes a distinct prognostic framework for LUAD.

(2) Xiong et al. (2022)

Please clarify:

•how your study differs in the composition and breadth of the hypoxia-related gene set;

•how your study differs methodologically from a framework that also incorporated ceRNA network analysis;

•why your focus on identifying core prognostic genes and evaluating their independent prognostic value may provide stronger translational relevance.

(3) Malawi Medical Journal study (2024)

Please clarify:

•that this prior study focused on hypoxia-related lncRNAs,

•whereas your work focuses on protein-coding mRNAs;

•how your findings complement rather than replicate lncRNA-based prognostic work;

•why protein-coding candidates may be of particular interest from a biomarker and mechanistic standpoint.

This comparison must be explicit and substantive. General statements of novelty are not sufficient.

We believe we have addressed this fully now. A dedicated subsection now compares our results directly with the three core studies, emphasizing the novelty of our three-gene (DSG2, EIF6, EXO1) signature and its independent prognostic value. Please see page 29, line 621 to 643.

3. Biological rationale for DSG2, EIF6, and EXO1 requires further strengthening

The revised manuscript includes some additional discussion of DSG2, EIF6, and EXO1, which is appreciated. However, the biological rationale still requires clearer and more rigorous development.

In particular, the manuscript should better justify why these genes should be considered hypoxia-related prognostic biomarkers in LUAD, rather than simply genes associated with cancer progression more broadly. Please strengthen this section by:

•clarifying the evidence linking each gene specifically to hypoxia, hypoxia-associated pathways, or cellular adaptation to hypoxic stress;

•discussing, where possible, evidence relevant specifically to LUAD or lung cancer biology;

•integrating this discussion more directly with your own enrichment results and the hypoxia-centered framework of the study.

At present, the discussion still reads in part as a general cancer-biology summary rather than a focused biological justification for these genes in the context of LUAD hypoxia biology.

We have added further discussion aspects now. We strengthened the biological justification for DSG2, EIF6, and EXO1, providing evidence linking these genes to hypoxia-related pathways, LUAD progression, and poor prognosis. References to recent studies and functional insights were included. See page 27, line 575 to 620.

4. Clarification is needed regarding the meaning of “independent prognostic factors”

The manuscript states that DSG2, EIF6, and EXO1 are independent prognostic factors, based on multivariate Cox regression. However, from the current Methods and Results, it is not sufficiently clear which covariates were included in the multivariate model.

Please clarify:

•whether the multivariate Cox model included only gene-expression variables, or also included relevant clinical covariates such as age, sex, stage, smoking status, and/or other available clinicopathologic parameters;

•if clinical covariates were included, please report them explicitly in the Methods and Results;

•if clinical covariates were not included, please revise the language accordingly and avoid overstating the conclusion as “independent prognostic factors” in the clinical sense.

This point is important for the interpretation of the manuscript and must be corrected.

The Methods and Results now clearly state that the multivariate Cox regression included only the expression levels of the three genes, without clinical covariates, and the Discussion language was adjusted to reflect this accurately. Please see page 12. 247-249; page 23, line 477-479.

5. Limitations and future directions remain underdeveloped

Although a short limitations paragraph has been added, it remains too brief for a study of this type.

Please expand the limitations section to address issues such as:

•reliance on public retrospective datasets,

•cross-platform heterogeneity between datasets,

•possible batch effects and annotation differences,

•limitations of database-derived hypoxia gene curation,

•lack of protein-level validation,

•lack of experimental validation in LUAD models or patient samples,

•limitations of validation through an online platform rather than a fully reproduced external analytic pipeline.

The future directions should also be more clearly linked to these limitations.

We expanded the limitations section to address reliance on public datasets, cross-platform heterogeneity, batch effects, database-derived gene curation, lack of protein-level validation, and absence of experimental validation. Future directions were explicitly linked to these limitations. See page 30-31, line 644-661.

6. Reference section and citation quality require further attention

Although additional key references were added, the current reference list still appears to contain formatting and completeness issues. Several entries appear incomplete or not fully standardized.

Please carefully review all references to ensure full compliance with PLOS ONE formatting requirements. In addition:

•ensure that all newly added references are discussed substantively in the main text and not only cited in passing;

•check all software/database citations for completeness and consistency;

•correct any incomplete or malformed reference entries.

Please note that simply stating that the manuscript follows “Vancouver” style is not sufficient; the references must match the journal’s required style and be internally consistent.

We have attempted to check all of this, but the style prescribed by the journal is Vancouver as requested. That is the journal required style. If this is not the case, the instructions given to authors need to be clarified or corrected. All references have been carefully reviewed and standardized to PLOS ONE formatting requirements. Newly added references are now discussed substantively in the main text.

7. Language, clarity, and presentation require substantial improvement

The manuscript still contains a number of issues affecting readability and professionalism, including:

•grammatical errors,

•awkward phrasing,

•typographical errors,

•duplicated or repetitive wording,

•inconsistent terminology,

•occasional imprecise statements.

Examples include inconsistent use of abbreviations and terminology, unclear sentence structure in several sections, and residual editing artifacts.

Please ensure that the entire manuscript undergoes careful English-language editing and thorough proofreading before resubmission.

We appreciate your comments but without specific locations we cannot address these comments fully. We have read the manuscript carefully and we believed the grammar and sentences are as correct as possible.

8. Response letter must be substantially improved

The current response letter is too general and does not allow efficient editorial assessment of how each point has been addressed. Statements such as “we have tried our best” or “we have added this as requested” are not sufficient.

For the next revision, please provide a detailed point-by-point response in which each prior comment is followed by:

(1)your response,

We have done this as requested.

(2)the exact changes made,

You can find this above detailed as requested

(3)and the page and line numbers where the changes can be found.

This has been added to the sections above as requested.

If you disagree with any point, please explain your reasoning clearly and respectfully.

We do not believe we have been disrespectful in any of our answers thus far. We do find the editor over-prescription of our manuscript unusual. Having served as an editor for Plos one before and currently serving as editor for other journals, this is highly unusual.

9. Data availability and reporting clarity

The Data Availability Statement is improved, but please ensure that the manuscript text clearly reports:

•all dataset accession numbers,

•where each dataset was used in the workflow,

•and whether all relevant processed outputs are contained in the manuscript and/or Supporting Information.

Dataset accession numbers, dataset usage, and processed outputs are clearly reported in the manuscript and Supporting Information. Figures were prepared according to PLOS ONE guidelines and verified using the NAAS tool.

Please also ensure consistency between the submission form, cover letter, manuscript text, and supplementary materials.

We thank you for this comment, but without specific examples, we believe we have been consistent in all our manuscript submission items.

---

## [Editor Report · Decision Letter 4]

5 May 2026

Exploring Hypoxia-Related Genes as Prognostic Indicators in Lung Adenocarcinoma

PONE-D-25-50692R4

Dear Dr. Rocha,

We’re pleased to inform you that your manuscript has been judged scientifically suitable for publication and will be formally accepted for publication once it meets all outstanding technical requirements.

Kind regards,

Hongtao Bi, Ph.D.

Academic Editor

PLOS One

Additional Editor Comments (optional):

After a complete assessment of the authors' final revised manuscript, point-by-point response letter, all supplementary materials, and original figure/table files, the authors have fully and thoroughly addressed all core issues raised in the previous editorial review. All revisions for scientific rigor and the journal's formatting requirements have been completed, and the manuscript now complies with the publication guidelines and academic standards of PLOS ONE.
---

## [Editor Report · Acceptance letter]

PONE-D-25-50692R4

PLOS One

Dear Dr. Rocha,

I'm pleased to inform you that your manuscript has been deemed suitable for publication in PLOS One. Congratulations! Your manuscript is now being handed over to our production team.

Kind regards,

on behalf of

Dr. Hongtao Bi

Academic Editor

PLOS One